# Occlusion-aware Non-Rigid Point Cloud Registration via Unsupervised Neural Deformation Correntropy

**Mingyang Zhao**[1], **Gaofeng Meng**[1,2,3,*] **& Dong-Ming Yan**[2,3] *
1. CAIR, Hong Kong Institute of Science & Innovation, Chinese Academy of Sciences
2. MAIS, Institute of Automation, Chinese Academy of Sciences
3. University of Chinese Academy of Sciences
https://github.com/zikai1/OAReg

## Abstract

Non-rigid alignment of point clouds is crucial for scene understanding, reconstruction, and various computer vision and robotics tasks. Recent advancements in implicit deformation networks for non-rigid registration have significantly reduced the reliance on large amounts of annotated training data. However, existing state-of-the-art methods still face challenges in handling *occlusion* scenarios. To address this issue, this paper introduces an innovative *unsupervised* method called **O**cclusion-**A**ware **R**egistration (**OAR**) for non-rigidly aligning point clouds. The key innovation of our method lies in the utilization of the adaptive correntropy function as a localized similarity measure, enabling us to treat individual points distinctly. In contrast to previous approaches that solely minimize overall deviations between two shapes, we combine unsupervised *implicit neural representations* with the *maximum correntropy criterion* to optimize the deformation of unoccluded regions. This effectively avoids collapsed, tearing, and other physically implausible results. Moreover, we present a theoretical analysis and establish the relationship between the maximum correntropy criterion and the commonly used Chamfer distance, highlighting that the correntropy-induced metric can be served as a more universal measure for point cloud analysis. Additionally, we introduce *locally linear reconstruction* to ensure that regions lacking correspondences between shapes still undergo physically natural deformations. Our method achieves superior or competitive performance compared to existing approaches, particularly when dealing with occluded geometries. We also demonstrate the versatility of our method in challenging tasks such as large deformations, shape interpolation, and shape completion under occlusion disturbances.

## 1 Introduction

Non-rigid point cloud registration is a critical and challenging problem within the domains of computer vision, robotics, and medical imaging. Its objective is to optimize a deformation field that enables precise alignment between geometric shapes. Due to its fundamental importance, this problem has diverse applications, including reconstructions (Slavcheva et al., 2017; Bozic et al., 2021; Park et al., 2021), generation (Barber et al., 2007; Wang et al., 2008), and animations (Chen et al., 2021; Siarohin et al., 2023).

Thanks to recent advancements in *neural representations* for 3D modeling and rendering (Park et al., 2019; Mescheder et al., 2019; Sitzmann et al., 2020; Mildenhall et al., 2020), coordinate-based networks (*i.e.*, utilizing 3D spatial positions as inputs) have made significant progress in fully automated non-rigid point cloud registration (Li et al., 2021a; Li & Harada, 2022b; Prokudin et al., 2023). Despite their impressive results on benchmark tests, the systematic exploration of coordinate-based networks for non-rigid point cloud registration involving *occlusion geometries* is comparatively limited and presents substantial challenges. Actually, the occlusion regime is quite relevant for practical

---

*Corresponding authors.

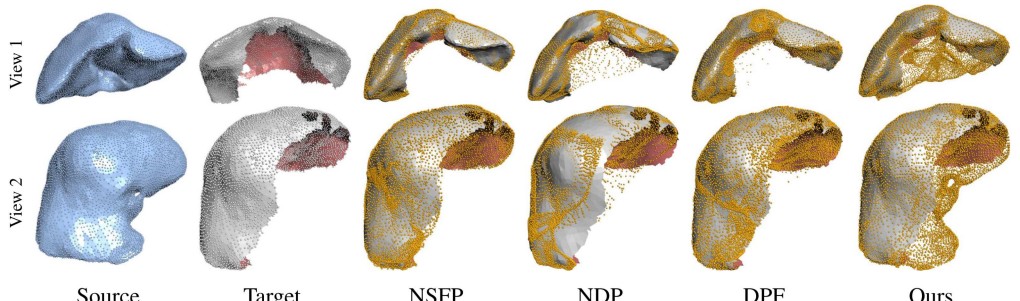

Figure 1: Non-rigid registration of point clouds under occlusion disturbances. The pre-operative liver (complete) and the intra-operative liver (occluded) serve as the source and target models, respectively. While competing approaches produce physically implausible results like collapses and tearing, our method achieves successful registrations (top) while faithfully preserving the physical structure (bottom) such as the blood vessel present in the source liver.

applications. For instance, in *Computer-Assisted Surgery* (CAS) scenarios, accurate registration of *complete* pre-operative CT or MRI models with *occluded* intra-operative scan data is crucial for surgical navigation, as the camera attached to the surgical instrument only provides a limited view of the abdomen during laparoscopic surgeries (Robu et al., 2018). It is surprising that current state-of-the-art algorithms struggle when dealing with scenarios where the source shape is complete and the target shape is occluded (see Fig. 1).

We analyze and argue that the unsatisfactory results can be attributed to the choice of the distance metric between two shapes. Previous approaches typically use the standard *Chamfer Distance* (CD) (Fan et al., 2017) as the similarity evaluation protocol (Li et al., 2021a; Li & Harada, 2022b; Prokudin et al., 2023), which directly finds the nearest point in the other point cloud and minimizes *holistic* deviations between shapes. While this is effective for registering two complete shapes, it can lead to *collapsed* or *physically implausible* results when dealing with a complete source shape and an occluded target geometry, as shown in Fig. 1.

To address the above issues, we present a novel and unsupervised non-rigid registration method for point clouds under occlusion. A key aspect of our method is that, instead of blindly optimizing the overall non-linear transformation, we introduce the use of the *Maximum Correntropy Criterion* (MCC) from statistical and information-theoretic learning (Erdogmus & Principe, 2002) to effectively handle occlusion. Correntropy serves as an *adaptive* and *local* similarity measure, allowing for differential treatment of individual points within the neural representations. Consequently, occluded points will make smaller contributions to the correntropy, enabling more precise and distinguished handling of occlusion. As demonstrated in subsequent experiments, this correntropy-induced metric effectively prevents collapses, tearing, and other physically implausible outcomes, in contrast to the commonly used CD metric.

Moreover, ensuring the physical plausibility of the deformation field for regions of the source geometry that lack correspondence in the target shape presents another significant challenge. This problem is considerably more complex than its rigid counterpart (Zhao et al., 2023), as it is inherently ill-posed and under-constrained. In non-rigid registration, there are numerous potential deformation solutions that can align with the partial deformations of the source surface. To mitigate this challenge, we draw inspiration from the concept of locally linear embedding in manifold learning (Roweis & Saul, 2000) and develop a robust geometric prior termed *Locally Linear Reconstruction* (LLR). The LLR approach formulates the problem as a *constrained non-linear least-squares optimization*, incorporating deformation regularity by linearly reconstructing the unknown out-of-part deformations using information from neighboring regions. As a result, LLR significantly enhances the naturalness of the deformation field and preserves higher-quality geometric details compared to previous approaches.

We conduct extensive experiments to demonstrate the superiority and competitiveness of the proposed method to baselines, including both traditional optimization-based and neural network-based approaches. This superiority is particularly pronounced in scenarios involving occlusion. Additionally, our method provides continuous deformation representations, enabling downstream applica-

tions in shape interpolation and shape completion. To summarize, the main technical contributions of this work are threefold as follows:

- We analyze the reasons why previous approaches fail to handle occlusion scenarios in non-rigid registration and introduce a novel unsupervised deformation framework to address this challenging problem.
- We pioneer the combination of maximum correntropy criterion with implicit neural representations to ensure physically plausible deformations and reveal its relationship to Chamfer distance.
- We formulate the deformation of regions lacking correspondence as a local linear reconstruction problem. This formulation can be solved through non-linear least-squares optimization and yields a closed-form solution.

## 2 RELATED WORK

We review the work that is closely related to ours. Readers are referred to (Tam et al., 2012; Deng et al., 2022) for more comprehensive studies.

**Non-rigid point cloud registration.** In contrast to shape matching, which aims to establish correct correspondences, non-rigid point cloud registration focuses on optimizing the deformation field. Amberg *et al.* (Amberg et al., 2007) extended the rigid *Iterative Closest Point* (ICP) algorithm (Besl & McKay, 1992) to non-rigid registration (NICP) by introducing stiffness parameters to control the deformation. *Coherent Point Drift* (CPD) represented point clouds using Gaussian mixture models (GMMs) and formulated the registration problem as a probabilistic density estimation process, while (Jian & Vemuri, 2010) directly minimized the Euclidean distance between two GMMs for registration. More recently, Hirose (Hirose, 2020; 2022) reformulated CPD in a Bayesian setting, improving CPD in scalability and robustness. Zhao et al. (2024) leveraged unsupervised clustering analysis to deal with non-rigid registration under large deformations. To enhance efficiency, deformation graph-based approaches (Sumner et al., 2007; Bozic et al., 2021; Zhao et al., 2022; Qin et al., 2023) represent shapes using sparsely sub-sampled node graphs, where deformation is associated with each graph node and applied to nearby geometry.

With the advent of deep learning, neural network-based approaches have also been developed for non-rigid point cloud registration. However, many of these methods rely on neural networks to extract features for point correspondences and subsequently employ classical registration methods like NICP for deformation optimization. For instance, SyNoRiM (Huang et al., 2022) employed 3D CNN networks to learn non-orthogonal functional bases for shape matching without relying on the Laplace-Beltrami operators (Ovsjanikov et al., 2012). Lepard (Li & Harada, 2022a) disentangled point cloud representations into feature and position spaces and developed a Transformer-based method for point-wise matching. DFR (Sun et al., 2024) conducted registration by aligning the source mesh towards a target point cloud using correspondences induced by deep functional maps (Litany et al., 2017). Cao & Bernard (2023) presented a self-supervised network for multimodal shape matching. While it shows promising performance, it does not address the deformations of occluded parts. Sundararaman et al. (2022) utilized an auto-decoder structure to implicitly align two volumes, which requires surface normals for training. Additionally, it employs a bi-directional Chamfer Distance for inferences, a metric that may be susceptible to occlusion. Unlike these approaches that primarily focus on shape matching and heavily depend on data annotations, our method is unsupervised, enabling us to achieve faithful registration that is more generalizable to unknown categories.

**Neural deformation representation.** The representation of deformation fields is a core aspect in non-rigid point cloud registration. Traditional methods often rely on manually defined deformation functions, such as the thin-plate spline (Bookstein, 1989) and the radial basis function (Yuille & Grzywacz, 1989). However, the emergence of implicit neural representations has introduced a new paradigm to describe deformations. Specifically, the coordinate-based *Multi-Layer Perception* (MLP) architecture utilizes an MLP to map the input coordinates to deformation fields, avoiding the need for explicit deformation definitions. For instance, NSFP (Li et al., 2021a) developed a coordinate-based MLP that implicitly regularized non-rigid deformations in scene flow estimation.

This method directly minimizes the Chamfer distance at runtime without relying on extensive labeled data to capture prior statistics. Building upon NSFP, NDP (Li & Harada, 2022b) extended the approach to a hierarchical motion decomposition using a pyramid structure. This enables controlling of the coarse-to-fine motion (from rigid to non-rigid) across low to high-frequency signals. More recently, DPF (Prokudin et al., 2023) introduced a method for modeling non-rigid surfaces also based on the Chamfer distance, where the deformation field was further regularized by the well-established as-isometric-as-possible (AIAP) (Kilian et al., 2007) constraint. We also utilize a coordinate-based MLP to represent the continuous deformation field, which allows us to model and parameterize the deformations in an unsupervised and flexible manner. However, unlike previous approaches that often face challenges in occlusion scenarios, we thoroughly analyze and address these issues, leading to higher-quality non-rigid point cloud registration results.

## 3 PRELIMINARIES

**Definition 1** (Santamaría et al. (2006); Liu et al. (2007)). *(Cross) correntropy $V$ is a generalized local similarity measure between two arbitrary scalar random variables $X$ and $Y$ defined as*

$$V(X, Y) = \mathbb{E}_{XY}[k(X, Y)] = \iint_{x,y} k(x, y) p_{XY}(x, y) dx dy, \quad (1)$$

*which is symmetric, positive, and bounded. Here $k(\cdot, \cdot)$ is a kernel function that satisfies the Mercer theorem (Vapnik, 2013), $\mathbb{E}$ denotes the expectation operator, and $p_{XY}$ is the joint probability density function over the space $(X, Y)$.*

In this work, we adopt the translation invariant Gaussian kernel $k_\sigma = \exp(-\frac{x^2}{2\sigma^2})$ to define the correntropy. This choice is motivated by the symmetric and positive definite nature of the Gaussian kernel, as well as its property of approaching zero as points move away from the center. Moreover, the kernel bandwidth $\sigma$ allows flexible control over the decaying factor and enables accurate registrations in the presence of various occlusion. We empirically investigate the influences of $\sigma$ on deformations in Appendix D.

**Definition 2** (Liu et al. (2007)). *Given the finite number of data points in the sample space, the correntropy-induced similarity metric (i.e., distance function) $\mathcal{M}$ between two vectors $\boldsymbol{x} = (x_1, x_2, \cdots, x_n) \in \mathbb{R}^n$ and $\boldsymbol{y} = (y_1, y_2, \cdots, y_n) \in \mathbb{R}^n$ is defined as*

$$\mathcal{M}(\boldsymbol{x}, \boldsymbol{y}) = (k_\sigma(0) - \frac{1}{n} \sum_{j=1}^{n} k_\sigma(x_j - y_j))^{\frac{1}{2}}. \quad (2)$$

In the machine learning community, for general optimization problems involving the unknown variable $\boldsymbol{\Theta}$, minimizing $\mathcal{M}(\boldsymbol{x}, \boldsymbol{y})$ is equivalent to maximizing the correntropy.

**Lemma 1.** *[Liu et al. (2007)] $\mathcal{M}(\boldsymbol{x}, \boldsymbol{y})$ is equivalent to the $\ell_2$ metric if $\boldsymbol{x}, \boldsymbol{y}$ are close, behaves similarly to the $\ell_1$ metric as $\boldsymbol{x}, \boldsymbol{y}$ get apart and approaches the $\ell_0$ when they are far apart.*

Therefore, Eq. (2) can be utilized as a novel cost function for *adaptive training*, referred to as the *Maximum Correntropy Criterion* (MCC). The pioneering work (He et al., 2010) utilizes MCC for face recognition. We explore the combination of unsupervised implicit neural representations and the MCC-induced local metric to address occlusion geometries in non-rigid point cloud registration. We also demonstrate the superior performance of our method over previous competing approaches.

## 4 PROPOSED METHOD

**Problem formulation.** Let $\mathbf{X} = \{\boldsymbol{x}_i \in \mathbb{R}^3\}_{i=1}^{M}$ and $\mathbf{Y} = \{\boldsymbol{y}_j \in \mathbb{R}^3\}_{j=1}^{N}$ denote two input point clouds, where $\mathbf{X}$ represents the target shape, possibly with occlusion, and $\mathbf{Y}$ is the source shape. Our objective is to solve the optimal non-linear map $\mathcal{T}_{\boldsymbol{\Theta}}$ parameterized by $\boldsymbol{\Theta}$ to minimize the shape deviation between the transformed source shape $\mathcal{T}_{\boldsymbol{\Theta}}(\mathbf{Y})$ and $\mathbf{X}$. Note that we also aim to preserve the physical plausibility of the deformation $\mathcal{T}_{\boldsymbol{\Theta}}$ regarding $\mathbf{Y}$, ensuring that issues encountered in previous approaches, such as collapsing or tearing of the geometric shape, are avoided.

### 4.1 Neural Deformation Correntropy

**Motivation.** Unlike previous approaches that directly minimize the *overall* difference between two shapes, we leverage the advantage of *local similarity* induced by correntropy. This allows us to adaptively differentiate between various parts of a shape, ensuring that occluded regions have minimal influence on the deformation process.

**Correntropy-induced metric.** We utilize the unsupervised neural presentations defined by a set of network parameters $\boldsymbol{\Theta}$ to implicitly parameterize the non-linear map $\mathcal{T}$. Based on MCC, our goal is to optimize the subsequent metric between $\mathbf{X} = \{\boldsymbol{x}_i \in \mathbb{R}^3\}_{i=1}^M$ and $\mathbf{Y} = \{\boldsymbol{y}_j \in \mathbb{R}^3\}_{j=1}^N$:

$$\min_{\boldsymbol{\Theta}} \mathcal{L}(\boldsymbol{\Theta}) = \mathbb{E}(\mathcal{M}(\boldsymbol{x}_i, \tilde{\boldsymbol{y}}_i)) + \mathbb{E}(\mathcal{M}(\mathcal{T}_{\boldsymbol{\Theta}}(\boldsymbol{y}_j), \tilde{\boldsymbol{x}}_j)), \tag{3}$$

where $\tilde{\boldsymbol{y}}_i = \min_{\boldsymbol{y}_j \in \mathcal{T}(\mathbf{Y})} \|\boldsymbol{x}_i - \mathcal{T}_{\boldsymbol{\Theta}}(\boldsymbol{y}_j)\|_2$, $\tilde{\boldsymbol{x}}_j = \min_{\boldsymbol{x}_i \in \mathbf{X}} \|\mathcal{T}_{\boldsymbol{\Theta}}(\boldsymbol{y}_j) - \boldsymbol{x}_i\|_2$. As a result, Eq. (3) assigns adaptive importance during network training, *i.e.*, giving significant importance to points with corresponding parts, while penalizing deformations in occluded regions. As demonstrated in subsequent experiments, our method effectively avoids physically implausible limitations, leading to more accurate and reasonable deformations.

**Relationship to Chamfer distance.** We further analyze the relationship between the maximum correntropy criterion and the Chamfer distance. Unlike CD that performs a holistic evaluation, MCC is a *local and adaptive* similarity metric. Based on Lemma 1, we can draw the following conclusion.

**Proposition 1.** *The Chamfer distance is a special case of Eq. (3), applicable only when both the source and target point clouds $\mathbf{X}$ and $\mathbf{Y}$ are complete and sufficiently close to each other. However, this equivalence does not hold when either of the inputs $\mathbf{X}$ and $\mathbf{Y}$ is occluded.*

Therefore, the correntropy-induced metric can serve as a more universal measure, potentially applicable to a wider range of point cloud analysis tasks.

### 4.2 Locally Linear Reconstruction

Neural deformation correntropy ensures physically plausible deformations for unoccluded parts through a local measure. However, addressing the challenge of making reasonable deformation of parts in the source shape that lack correspondence in the target shape remains highly ill-posed. To tackle this, we propose incorporating a geometric prior by modeling the deformation field of occluded parts through *Locally Linear Reconstruction* (LLR), which transforms the unconstrained problem into a *constrained non-linear optimization*. This allows us not only maintaining a smooth and coherent deformation field for unoccluded parts but also ensuring *physically plausible results* for parts that lack correspondence.

Inspired by the principle of locally linear embedding in nonlinear spectral dimensionality reduction (Roweis & Saul, 2000), we develop a novel method to preserve the linear pattern of the *local geometric structure* and reconstruct each point in the *deformation space*, especially for points that lack correspondences, by utilizing its neighboring structure. Concretely, for each source point $\boldsymbol{y}_j \in \mathbb{R}^3$ in $\mathbf{Y}$, we formulate the locally linear reconstruction process as the subsequent optimization problem:

$$\min_{\boldsymbol{w}_j} \mathcal{L}(\boldsymbol{w}_j) = \frac{1}{2} \|\boldsymbol{y}_j - \mathbf{Z}_j \boldsymbol{w}_j\|_2^2 = \frac{1}{2} \boldsymbol{w}_j^\top \mathbf{G}_j \boldsymbol{w}_j, \text{ s.t. } \boldsymbol{w}_j^\top \mathbf{1}_k = 1, \tag{4}$$

where $\boldsymbol{w}_j = [w_{j1}, w_{j2}, \cdots, w_{jk}]^T \in \mathbb{R}^k$ is the set of unknown weight factors, $\mathbf{Z}_j = [\boldsymbol{z}_{j1}, \boldsymbol{z}_{j2}, \cdots, \boldsymbol{z}_{jk}] \in \mathbb{R}^{3 \times k}$ is the $k$-nearest neighbors of $\boldsymbol{y}_j$ (the investigation of the influences of $k$ on deformations is presented in Appendix D), $\mathbf{G}_j = (\boldsymbol{y}_j \mathbf{1}^\top - \mathbf{Z}_j)^\top (\boldsymbol{y}_j \mathbf{1}^\top - \mathbf{Z}_j) \in \mathbb{R}^{k \times k}$ is the Gram matrix, $\mathbf{1}_k \in \mathbb{R}^k$ is the $k$-dimensional vector of all ones, and $\top$ represents the matrix transpose operator.

Eq. (4) is a constrained non-linear least-squares regression problem, utilizing the Lagrange multiplier, we obtain the *closed-form solution* of $\boldsymbol{w}_j$ for each $\boldsymbol{y}_j$ (derivations are presented in Appendix B)

$$\boldsymbol{w}_j = \lambda_j \mathbf{G}_j^{-1} \mathbf{1}_k = \frac{\mathbf{G}_j^{-1} \mathbf{1}_k}{\mathbf{1}_k^\top \mathbf{G}_j^{-1} \mathbf{1}_k} \tag{5}$$

with the corresponding Lagrange multiplier

$$\lambda_j = \frac{1}{\mathbf{1}_k^\top \mathbf{G}_j^{-1} \mathbf{1}_k}. \tag{6}$$

**LLR regularization.**   Once obtaining the weight vectors $\{\boldsymbol{w}_j \in \mathbb{R}^k\}_{j=1}^N$ for each $\boldsymbol{y}_j \in \mathbf{Y}$, we keep them fixed. Then, we utilize these weight vectors to *regularize* the displacement field, aiming to promote its smoothness and ensure physical plausibility in the deformation space throughout the entire optimization process even without correspondences, *i.e.*,

$$\mathcal{R}(\boldsymbol{\Theta}) = \sum_{j=1}^N \|\mathcal{T}_{\boldsymbol{\Theta}}(\boldsymbol{y}_j) - \mathcal{T}_{\boldsymbol{\Theta}}(\mathbf{Z}_j)^\top \boldsymbol{w}_j\|_2. \tag{7}$$

**Geometric meaning of LLR.**   Unlike rigid registration, where each point shares the same spatial transformation (*i.e.*, rotation and translation), non-rigid deformation requires solving for each point's individual displacement. In challenging occlusion scenarios, Eq. (7) demonstrates that the deformation of occluded parts can be inferred from, or propagated through, the deformations of neighboring points via locally linear reconstruction. Fig. 2 presents a schematic description of the geometric meaning.

## 4.3 UNSUPERVISED OPTIMIZATION

Given the active research and powerful fitting ability of implicit neural representations across various domains, we adopt a coordinate-based MLP with periodic activations (Sitzmann et al., 2020) as our deformation network $\mathcal{T}$. Our objective is to optimize a set of network parameters $\boldsymbol{\Theta}$ to define the deformation space. Additionally, as a test-time optimization method, our method is unsupervised, requiring no data annotations. The neural network is composed of three hidden layers, each consisting of 128 neurons. We obtain the optimal neural network parameters $\boldsymbol{\Theta}^*$ by optimizing the following cost function with automatic differentiation:

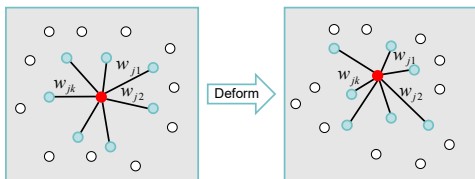

Figure 2: Geometric meaning of locally linear reconstruction. Left: The initial source shape along with the reconstruction weight vector $\boldsymbol{w}_j$. Right: The deformed shape reconstructed using the same $\boldsymbol{w}_j$.

$$\boldsymbol{\Theta}^* = \arg\min_{\boldsymbol{\Theta}} \mathcal{F}(\boldsymbol{\Theta}) = \alpha_1 \mathcal{L}(\boldsymbol{\Theta}) + \alpha_2 \mathcal{R}(\boldsymbol{\Theta}), \tag{8}$$

where $\alpha_1, \alpha_2 \in \mathbb{R}^+$ are trade-off factors. Finally, the resulting deformed point cloud $\mathcal{T}_{\boldsymbol{\Theta}^*}(\mathbf{Y})$ is obtained by applying the learned deformation network $\mathcal{T}_{\boldsymbol{\Theta}^*}$ to the source model $\mathbf{Y}$.

## 5 EXPERIMENTAL RESULTS

We conduct comprehensive experiments to validate the performance of our method and compare it with state-of-the-art approaches in both occluded and complete settings. Additionally, we showcase the versatility of our method by applying it to *shape interpolation* and *shape completion* tasks.

**Implementation details.**   Our method is implemented using the PyTorch framework (Paszke et al., 2019), and we employ the Adam optimizer (Kingma & Ba, 2014) with default settings $\beta_1 = 0.9$, $\beta_2 = 0.999$, and $\epsilon = 10^{-8}$ to optimize the objective function defined in Eq. (8). The initial learning rate is set to $lr = 1e-4$, and the network is trained for 200 epochs with a batch size of 1. To dynamically update the learning rate, we utilize the `ReduceLROnPlateau` technique with a patience value of 1. During optimization, we fix $\alpha_1 = 10^4$ and $\alpha_2 = 10^2$, which yield appealing results across various scenarios. All experiments are conducted on a machine equipped with an NVIDIA A30 GPU and an Intel(R) Xeon(R) Gold 6248R CPU @ 3.00GHz. We utilize publicly available implementations of baseline approaches for evaluation, with their parameters either fine-tuned by ourselves or suggested by the authors to obtain their best results.

Table 1: Quantitative comparisons on the occluded Open-CAS liver dataset. ↑ means larger values are better while ↓ means smaller values are better. **Bond** fonts indicate the top performer.

| Metric | Liver 1 | | | | Liver 2 | | | | Liver 3 | | | |
|---|---|---|---|---|---|---|---|---|---|---|---|---|
| Method | EPE ↓ | AccS ↑ | AccR ↑ | Outlier ↓ | EPE ↓ | AccS ↑ | AccR ↑ | Outlier ↓ | EPE ↓ | AccS ↑ | AccR ↑ | Outlier ↓ |
| GBCPD (Hirose, 2022) | 11.849 | 18.770 | 65.263 | 0.000 | 9.169 | 48.871 | 75.534 | 0.000 | 16.240 | 31.770 | 53.162 | 0.000 |
| AMM_NRR (Yao et al., 2023) | 28.505 | 21.549 | 36.683 | 1.353 | 19.188 | 36.997 | 47.030 | 0.000 | 18.744 | 35.593 | 49.289 | 0.000 |
| NSFP (Li et al., 2021a) | 49.655 | 18.057 | 39.494 | 27.218 | 32.520 | 45.011 | 53.659 | 9.514 | 28.789 | 37.742 | 50.696 | 14.167 |
| NDP (Li & Harada, 2022b) | 58.470 | 2.795 | 8.504 | 25.961 | 37.456 | 10.357 | 21.674 | 11.131 | 29.508 | 26.675 | 43.407 | 12.780 |
| DPF (Prokudin et al., 2023) | 35.090 | 9.065 | 21.797 | 1.225 | 25.180 | 20.158 | 46.976 | 0.000 | 23.045 | 26.507 | 43.491 | 5.041 |
| OAR (Ours) | **8.662** | **29.228** | **96.813** | **0.000** | **5.687** | **75.193** | **97.184** | **0.000** | **12.112** | **42.372** | **56.564** | **0.000** |

**Evaluation metrics.** To quantitatively evaluate the results of non-rigid deformation, we adopt four metrics described in (Li & Harada, 2022b). 1) *End-Point Error* (EPE): This metric calculates the average norm of the 3D deformation error vectors across all points; 2) *3D Accuracy Strict* (AccS): AccS measures the percentage of points whose relative error is less than $2.5\%$ or 2.5 cm. 3) *3D Accuracy Relaxed* (AccR): AccR measures the percentage of points whose relative error is less than $5\%$ or 5 cm; 4) *Outlier Ratio*: The outlier ratio calculates the percentage of points whose relative error is greater than $30\%$. Concrete definitions of these metrics are presented in Appendix A. Note that **AccS** and **AccR** are considered the most important metrics as they directly assess the ratio of accurately registered points, providing a reliable assessment of the registration quality (Li & Harada, 2022b).

## 5.1 COMPARISONS

We first assess the proposed algorithm on occlusion scenarios using the silico liver dataset from Open-CAS (ope). Open-CAS is an open collection of datasets specifically designed for validating *computer-assisted surgery* systems. The silico liver dataset consists of three pairs of liver models that have undergone deformation using a non-linear biomechanical model. Each pair, as illustrated in Fig. 1, consists of a source complete liver model and a target occluded liver model, along with its corresponding ground truth complete model. These pairs represent the pre-operative and intra-operative shapes, respectively. We adopt state-of-the-art approaches for comparison, including both traditional optimization-based methods GBCPD (Hirose, 2022), AMM_NRR (Yao et al., 2023) as well as neural network-based competitors NSFP (Li et al., 2021a), NDP (Li & Harada, 2022b), and DPF (Prokudin et al., 2023).

The quantitative comparison results presented in Tab. 1 demonstrate that our method consistently outperforms competing approaches with higher registration accuracy across all livers, indicating its robustness and stability to align the geometric shapes under occlusion disturbances. Notably, our method achieves the highest AccS and AccR values in all tests, surpassing competitors by a significant margin. Additionally, our method delivers the lowest EPE values among all approaches and maintains a zero outlier ratio in all tests. These achievements highlight the success of our method in avoiding significant registration errors such as collapses and tearing, thereby ensuring a high level of accuracy. The qualitative comparison presented in Fig. 1 further illustrates the advantage of our method in preserving *detailed physical structures* such as the blood vessel of the source liver.

## 5.2 REGISTRATION UNDER DIFFERENT TYPES OF OCCLUSION

To further investigate the performance of our proposed method in handling occlusion, we create three occlusion datasets using three different shapes (cat, horse, and dog) from the TOSCA dataset (Bronstein et al., 2008). Each dataset consists of five types of occlusion, such as the occlusion present in the body of the cat and the tail of the dog, as shown in Fig. 3. We exclude the outlier ratio from our analysis since the deformation solution is not unique in these cases. Additionally, we exclude AMM_NRR from our analysis due to its significant deviations from the target poses, which render the error metrics unreliable. The average metrics of all tests are reported in Tab. 2. The results indicate that our method still achieves the highest registration accuracy across different occlusion scenarios, especially in terms of the AccR metric. Although our method is slightly slower than other neural networks-based approaches, it is much faster than traditional optimization-based GBCPD. Several qualitative comparison results are presented in Fig. 3. As observed, our method success-

Table 2: Quantitative comparisons on different types of occlusion. ↑ means larger values are better while ↓ means smaller values are better. **Bond** fonts indicate the top performer.

| Metric / Method | Cat EPE ↓ | Cat AccS ↑ | Cat AccR ↑ | Horse EPE ↓ | Horse AccS ↑ | Horse AccR ↑ | Dog EPE ↓ | Dog AccS ↑ | Dog AccR ↑ | Time (s) |
|---|---|---|---|---|---|---|---|---|---|---|
| GBCPD (Hirose, 2022) | 4.118 | 17.123 | 41.104 | 3.273 | 38.353 | 52.435 | 4.359 | 54.472 | 77.375 | 185.621 |
| NSFP (Li et al., 2021a) | 4.591 | 13.617 | 40.397 | 2.761 | 25.484 | 53.698 | 4.159 | 61.486 | 78.940 | 4.576 |
| NDP (Li & Harada, 2022b) | 7.280 | 13.401 | 32.340 | 4.147 | 30.922 | 48.070 | 11.141 | 14.710 | 42.522 | 5.858 |
| DPF (Prokudin et al., 2023) | 2.688 | 50.048 | 75.350 | 2.885 | 27.819 | 55.056 | 3.888 | 64.480 | 80.814 | 9.518 |
| OAR (Ours) | **2.559** | **50.569** | **77.171** | **2.249** | **40.046** | **65.317** | **2.467** | **76.986** | **89.779** | 13.646 |

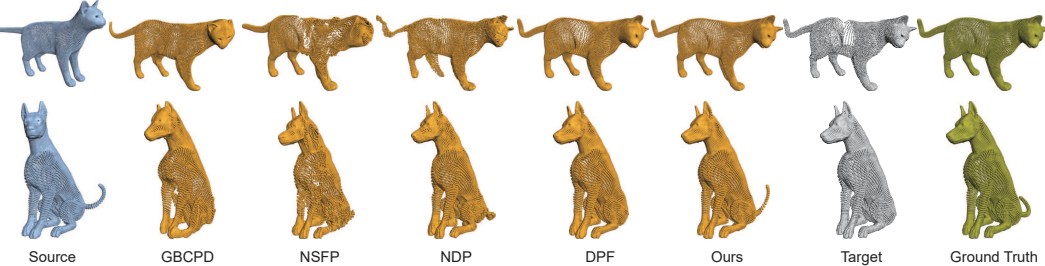

Source     GBCPD     NSFP     NDP     DPF     Ours     Target     Ground Truth

Figure 3: Qualitative comparison on different types of occlusion. We register the complete source cat and dog model to the target shape with **body** and **tail** occlusion, respectively.

fully retains the dog tail without any collapse. More comparison results are presented in Fig. 10 and Fig. 11 of the *Appendix*.

## 5.3 REGISTRATION WITH CORRESPONDENCE

In addition to handling occlusion scenarios, we also demonstrate the capability of the proposed method for general large deformation registration tasks. To this end, we combine the pre-trained geometric feature descriptor Lepard (Li & Harada, 2022a) with our method for non-rigid registration on the 4DMatch and 4DLoMatch animation benchmark datasets (Li & Harada, 2022a) (details are presented in Appendix I). These datasets are collected from the DeformingThings4D sequence (Li et al., 2021b), which contains challenging scenarios with partial overlap, occlusion, and large motions. The 4DMatch dataset comprises point clouds with overlap ratios ranging from $45\%$ to $92\%$, while the 4DLoMatch dataset exhibits lower overlap ratios of $15\% - 45\%$. Following (Li & Harada, 2022b), we adopt the filtered dataset that has undergone the removal of near-rigid transformations for testing. Specifically, the 4DMatch dataset consists of 2,590 pairs of test, while the 4DLoMatch dataset contains 1,175 pairs.

Tab. 3 summarizes the quantitative comparison results, with the exception of our method, where the remaining results are taken from (Li & Harada, 2022b). It is important to note that all supervised models are re-trained on 4DMatch's training split prior to evaluation. Although our method is unsupervised, it still consistently achieves superior or competitive results compared to the competitors, particularly in terms of the two most important metrics, AccS and AccR. Moreover, it is worth noting that our method also delivers competitive performance on the highly challenging 4DLoMatch dataset even without any fine-tuning. Qualitative results are presented in Fig. 13 of the *Appendix*.

## 5.4 HUMAN REGISTRATION ON OCCLUDED DEPTH VIEWS

We further evaluate the proposed method for registering human point clouds captured from RGB-D cameras, which can be used for non-rigid reconstruction. As shown in Fig. 16, the sequences from (Innmann et al., 2016) exhibit occlusion disturbances due to human motion. The qualitative results presented in Fig. 17 evidence that our method still achieves high-quality deformations despite these challenges. In contrast to DPF, which produces physically unreasonable deformations such as collapse and distortion of body parts, our method consistently preserves realistic deformations, particularly evident in the human hands and arms.

Table 3: Quantitative comparisons on the 4DMatch and 4DLoMatch benchmark datasets. The best and second best performers of AccS and AccR are highlighted in **bold** and underlined, respectively.

| Metric | 4DMatch | | | | 4DLoMatch | | | |
|---|---|---|---|---|---|---|---|---|
| Method | EPE ↓ | AccS ↑ | AccR ↑ | Outlier ↓ | EPE ↓ | AccS ↑ | AccR ↑ | Outlier ↓ |
| Lepard (Li & Harada, 2022a)+SVD | 0.137 | 6.91 | 24.50 | 43.43 | 0.160 | 5.27 | 19.77 | 44.16 |
| PointPWC (Wu et al., 2019) | 0.182 | 6.25 | 21.49 | 52.07 | 0.279 | 1.69 | 8.15 | 55.70 |
| FLOT (Puy et al., 2020) | 0.133 | 7.66 | 27.15 | 40.49 | 0.210 | 2.73 | 13.08 | 42.51 |
| GeomFmaps (Donati et al., 2020) | 0.152 | 12.34 | 32.56 | 37.90 | 0.148 | 1.85 | 6.51 | 64.63 |
| SyNoRiM-pw (Huang et al., 2022) | 0.099 | 22.91 | 49.86 | 26.01 | 0.170 | 10.55 | 30.17 | 31.12 |
| Lepard+NICP (Li & Harada, 2022a) | 0.097 | 51.93 | 65.32 | 23.02 | 0.283 | 16.80 | 26.39 | 52.99 |
| Lepard+NDP (Li & Harada, 2022b) | 0.077 | **61.30** | 74.12 | 17.37 | 0.177 | 26.59 | 41.05 | 33.81 |
| Lepard+OAR (Ours) | 0.059 | 59.32 | **74.33** | 16.41 | 0.251 | **27.25** | **42.01** | 45.04 |

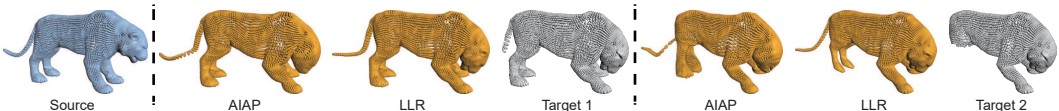

Source    AIAP    LLR    Target 1    AIAP    LLR    Target 2

Figure 4: Qualitative comparison of AIAP and LLR with increasing occlusion. As observed, LLR enables more natural registration results and maintains geometric details such as the facial expression more faithfully.

## 5.5 ROBUSTNESS TEST

We also assess the robustness of our developed method against noise and outlier disturbances. To this end, we introduce Gaussian noise with increasing intensity and varying outlier ratios (%)

| Intensity/Ratio | 0.1 | 0.2 | 0.3 | 0.4 | 0.5 | 0.6 |
|---|---|---|---|---|---|---|
| Noise | 88.712 | 89.104 | 89.678 | 85.701 | 84.395 | 81.807 |
| Outlier | 84.122 | 88.017 | 86.719 | 89.074 | 86.171 | 83.999 |

to the point clouds. The inset table reports the statistical results, while Fig. 19 and Fig. 20 present the qualitative outcomes. As observed, our method demonstrates strong resilience against both noise and outliers, maintaining performance even with noise levels up to $0.6\%$ and in the contamination of $2,350$ outliers.

## 5.6 ABLATION STUDY

**The effect of LLR.** To demonstrate the effect of the proposed LLR for non-rigid point cloud registration under occlusion disturbances, we construct a series of models with progressively increasing levels of occlusion. In Fig. 4, we illustrate this by manually removing some parts of the lion model, starting from the tail and extending towards the body. The statistical results in Fig. 5 indicate that our method equipped with the LLR constraint (*i.e.*, MCC+LLR), which helps preserve the local neighboring geometry, consistently produces higher-quality registration results compared to the single MCC framework.

**AIAP VS. LLR.** We also compare the performance of the commonly used AIAP regularization with the proposed LLR in handling challenging occlusion scenarios. As observed in Fig. 5, MCC+AIAP delivers similar results to MCC, while MCC+LLR consistently outperforms MCC+AIAP across different levels of occlusion. Moreover, benefiting from the reconstruction scheme and maintaining the local structure, LLR excels at preserving essential geometric details,

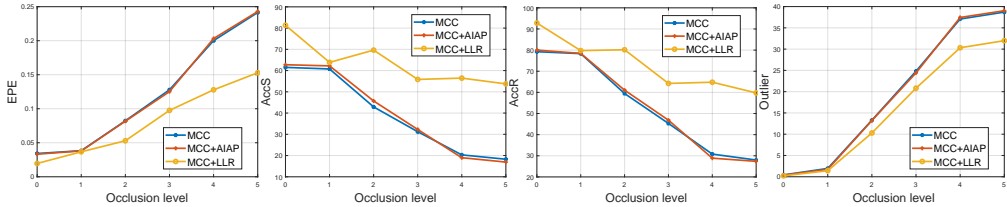

Figure 5: Ablation study of the LLR effect and the comparison between AIAP and LLR manners.

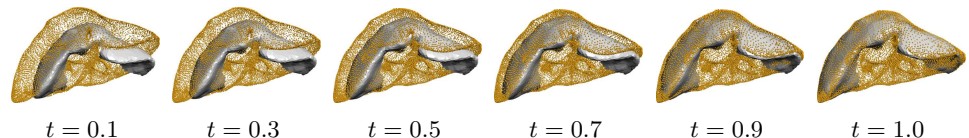

$t = 0.1$     $t = 0.3$     $t = 0.5$     $t = 0.7$     $t = 0.9$     $t = 1.0$

Figure 6: Application of the proposed non-rigid registration method to shape interpolation.

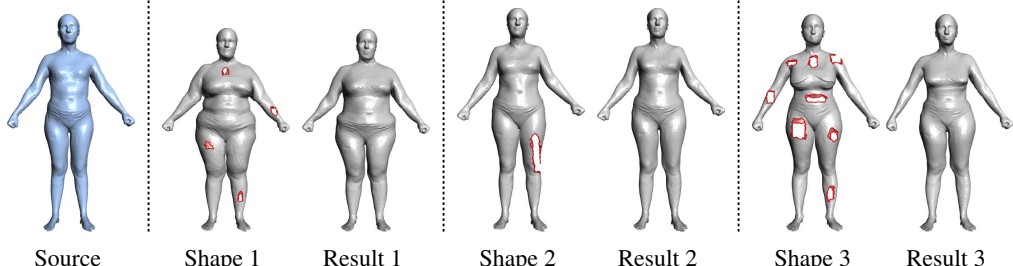

Source     Shape 1     Result 1     Shape 2     Result 2     Shape 3     Result 3

Figure 7: Application of the proposed non-rigid registration method to mesh hole filling.

such as the intricate facial features of the lion in Fig. 4, after non-rigid deformation. This stands in contrast to AIAP under occlusion disturbances, which struggles to maintain such details.

## 5.7 APPLICATIONS

**Shape interpolation.** Since our method successfully avoids physically implausible scenarios, such as collapses and tearing, and employs an unsupervised neural implicit representation, it inherently allows for continuous deformation. In Fig. 6, we illustrate the shape interpolation effect using our non-rigid deformation under occlusion, where we vary the timestamp $t$ within the range of $[0.1, 1.0]$. For instance, the interpolation shape with $t = 0.5$ is represented by $\mathbf{Y}_{t=0.5} = \mathbf{Y} + t * \nu(\mathbf{Y}) = \mathbf{Y} + 0.5 * \nu(\mathbf{Y})$ with $\nu$ denoting the learned displacement field. As observed, our method not only achieves accurate registration at $t = 1.0$ but also maintains substantially smooth and physically feasible interpolation configurations throughout the entire deformation process.

**Shape completion.** We also apply our method to address the problem of shape completion. In Fig. 7, we utilize the source mesh model (Yang et al., 2014) as the standard template, while the other shapes exhibit various types of holes caused by occlusion, serving as the targets. It is evident that by deforming the template model to align with the target shapes and leveraging the highly accurate outcomes, our method offers a feasible solution to the ill-posed mesh hole filling task without requiring any additional annotations and constraints. More hole filling results are reported in Fig. 14 of the *Appendix*.

## 6 CONCLUSIONS

We introduced a novel unsupervised solution to address the challenging issue of occlusion disturbances in non-rigid point cloud registration, a problem that has not received adequate attention in prior approaches. We provide a comprehensive analysis to elucidate the reasons why previous methods struggle with occlusion scenarios. The first contribution of our method is maximizing the local similarity between source and target geometries, accomplished through the utilization of the adaptive maximization correntropy criterion. We also establish a connection between the correntropy-induced metric and the commonly used Chamfer distance, highlighting that this correntropy-induced metric can serve as a more general measure. Additionally, we develop a locally linear reconstruction paradigm to ensure that deformations in the occluded regions are physically plausible while preserving intricate geometric details of deformed shapes, in contrast to the commonly used AIAP metric. Our method achieves higher-quality non-rigid registration results across a range of datasets, while effectively mitigating the issues of collapses and tearing that are encountered in previous approaches. Moreover, our method ensures continuous deformation, allowing for promising applications in shape interpolation and shape completion tasks.

## ACKNOWLEDGEMENTS

We sincerely thank the anonymous reviewers for their helpful comments which improved the quality of the paper. This work is partially funded by the National Natural Science Foundation of China (12494553, 12494550, 62172415, and 62376267), the Strategic Priority Research Program of the Chinese Academy of Sciences (XDB0640200 and XDB0640000), the Beijing Natural Science Foundation (Z240002), and the innoHK project.

## REPRODUCIBILITY STATEMENT

The main contribution of this work is the development of an unsupervised neural deformation method that combines correntropy-induced metric and locally linear reconstruction with implicit neural representations to ensure physically reasonable deformations of occluded point clouds. To reproduce the experimental results, we elaborate on the data pre-processing, metric definitions, and hyper-parameter settings in Appendix A and Appendix D. To facilitate reproducibility, we also publicly release our code.

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

APPENDIX

In this appendix, we provide additional content to support our paper. Concretely, we present the data pre-processing step of normalization and the definition of used evaluation metrics in Appendix A, followed by the theoretical derivation of the closed-form weight solution in Appendix B. Appendix C presents more details and analyses of Lemma 1. Appendix D investigates the influences of the number of nearest neighbors $k$ for each $\boldsymbol{y}_j \in \mathbf{Y}$ and the kernel bandwidth $\sigma^2$ on registration under occlusion disturbances. We present further ablation studies to investigate the regularization effect of implicit neural networks and activation functions in Appendix E. The differences and connections of MCC with robust functions are presented in Appendix F. Additional ablation studies of MCC and Chamfer distance as well as its variants are reported in Appendix G and Appendix H, separately. Appendix I gives the details of how to integrate the geometric descriptors with our proposed optimization framework. Then, we discuss the limitations and future work of the proposed algorithm in Appendix J. Finally, in Appendix K, we present more quantitative and qualitative results. These results include the registration on various types of occlusion from the TOSCA dataset, liver dataset, 4DMatch and 4DLoMatch datasets, shape completion for the challenging mesh hole filling task, human RGB-D views, failure cases, as well as robustness test.

## A  DATA PRE-PROCESSING AND THE METRIC DEFINITION

Given a pair of input point clouds $\mathbf{X} = \{\boldsymbol{x}_i \in \mathbb{R}^3\}_{i=1}^M$ and $\mathbf{Y} = \{\boldsymbol{y}_j \in \mathbb{R}^3\}_{j=1}^N$, where $\mathbf{X}$ and $\mathbf{Y}$ represent the target and source point clouds, separately, we first normalize them to follow a normal distribution, *i.e.*,

$$\mathcal{N}(\mathbf{X}) = (\mathbf{X} - \mathbf{1}_M \mu_{\mathbf{X}})/\sigma_{\mathbf{X}}, \tag{9}$$

$$\mathcal{N}(\mathbf{Y}) = (\mathbf{Y} - \mathbf{1}_N \mu_{\mathbf{Y}})/\sigma_{\mathbf{Y}}, \tag{10}$$

where $\mu_* \in \mathbb{R}^3$ represents the mean point in the set of $*$ and $\sigma_* \in \mathbb{R}^+$ is the farthest Euclidean distance from $\mu_*$ to the elements in $*$. $\mathbf{1}_M \in \mathbb{R}^M$ and $\mathbf{1}_N \in \mathbb{R}^N$ are the vectors of all ones. However, when evaluating the registration results, we still use the original inputs by first performing denormalization.

Specifically, we compute the deviations between the deformed shape $\widehat{\mathbf{Y}} = \sigma_{\mathbf{X}} \mathcal{T}(\mathcal{N}(\mathbf{Y})) + \mathbf{1}_N \mu_{\mathbf{X}}$ and the complete shape $\widehat{\mathbf{X}}$ of $\mathbf{X}$ in terms of the metrics EnE, AccS, AccR, and Outlier Ratio by

$$\mathrm{EPE}(\widehat{\mathbf{Y}}, \widehat{\mathbf{X}}) = \frac{1}{N} \sqrt{\mathrm{Tr}\{(\widehat{\mathbf{Y}} - \widehat{\mathbf{X}})^\top (\widehat{\mathbf{Y}} - \widehat{\mathbf{X}})\}},$$

$$\mathrm{AccS}(\widehat{\mathbf{Y}}, \widehat{\mathbf{X}}) = \frac{1}{N} \sum_{j=1}^N \mathbb{I}(\|\widehat{\boldsymbol{y}}_j - \widehat{\boldsymbol{x}}_j\|_2 < 0.025) \times 100\%,$$

$$\mathrm{AccR}(\widehat{\mathbf{Y}}, \widehat{\mathbf{X}}) = \frac{1}{N} \sum_{j=1}^N \mathbb{I}(\|\widehat{\boldsymbol{y}}_j - \widehat{\boldsymbol{x}}_j\|_2 < 0.05) \times 100\%, \tag{11}$$

$$\mathrm{Outlier}(\widehat{\mathbf{Y}}, \widehat{\mathbf{X}}) = \frac{1}{N} \sum_{j=1}^N \mathbb{I}(\|\widehat{\boldsymbol{y}}_j - \widehat{\boldsymbol{x}}_j\|_2 > 0.3) \times 100\%,$$

where Tr and $\top$ are the trace operator and transpose operator of a matrix, separately. $\mathbb{I}$ is the indicator function and $\|\|_2$ represents the Euclidean distance.

## B  DERIVATION OF THE CLOSED-FORM SOLUTION IN EQ. (5) OF THE PAPER

To make the proposed method self-contained, we present the derivation of the closed-form solution in Eq. (5) as follows. According to the Lagrange multiplier, we have

$$\mathcal{L}(\boldsymbol{w}_j, \lambda_j) = \frac{1}{2} \boldsymbol{w}_j^\top \mathbf{G}_j \boldsymbol{w}_j - \lambda_j (\boldsymbol{w}_j^\top \mathbf{1}_k - 1). \tag{12}$$

By equating the partial derivatives of $\mathcal{L}(\boldsymbol{w}_j, \lambda_j)$ to zero, we obtain

$$
\frac{\partial \mathcal{L}}{\partial \boldsymbol{w}_j} = \boldsymbol{G}_j \boldsymbol{w}_j - \lambda_j \mathbf{1}_k = \mathbf{0},
$$
$$
\implies \boldsymbol{w}_j = \boldsymbol{G}_j^{-1} \lambda_j \mathbf{1}_k = \lambda_j \boldsymbol{G}_j^{-1} \mathbf{1}_k. \tag{13}
$$
$$
\frac{\partial \mathcal{L}}{\partial \lambda_j} = \boldsymbol{w}_j^\top \mathbf{1}_k - 1 = 0 \implies \boldsymbol{w}_j^\top \mathbf{1}_k = 1.
$$

Then, we have

$$
\lambda_j = \frac{1}{\mathbf{1}_k^\top \boldsymbol{G}_j^{\top -1} \mathbf{1}_k^\top} = \frac{1}{\mathbf{1}_k^\top \boldsymbol{G}_j^{-1} \mathbf{1}_k^\top}. \tag{14}
$$

Thus,

$$
\boldsymbol{w}_j = \frac{\boldsymbol{G}_j^{-1} \mathbf{1}_k}{\mathbf{1}_k^\top \boldsymbol{G}_j^{-1} \mathbf{1}_k^\top}. \tag{15}
$$

This completes the derivation of Eq. (5) and Eq. (6).

## C   MORE DETAILS AND ANALYSES OF LEMMA 1

We present more details of Lemma 1. We can rewrite the correntropy of Gaussian kernel as

$$
V_\sigma(X, Y) = 1/(\sqrt{2\pi}\sigma) \sum_{n=0}^{\infty} \left((-1)^n\right) / (2^n n!) \, \mathbb{E}\left[ \left((X-Y)^{2n}\right) / \left(\sigma^{2n}\right) \right], \tag{16}
$$

which involves all the even moments of the random variable $X - Y$. When two points are in close proximity, the second-order moment tends to be dominant, causing correntropy to approach correlation, as the higher-order moments decay more rapidly. As the points become more distant, as shown in (Liu et al., 2007), correntropy transitions from behaving like an $\ell_1$ norm to resembling an $\ell_0$ norm. This transition further illustrates the inherent robustness of the MCC metric.

## D   INVESTIGATION OF $k$ AND $\sigma^2$ ON DEFORMATION

We adopt the occluded liver 1 dataset (ope) and vary the number of neighbors $k \in [2, 30]$ with $\Delta k = 2$ and $\sigma^2 \in [0.2, 2]$ with $\Delta \sigma^2 = 0.2$ to investigate their influences on non-rigid registration under occlusion disturbances. As shown in Fig. 8, our method consistently achieves significantly lower outlier ratios across different values of $k$ and $\sigma^2$. With increasing values of $k$, our method not only demonstrates higher AccS and AccR results but also with slower EPE. Nevertheless, larger values of $k$ typically result in higher computational complexity due to the matrix inverse operation in Eq. (5) of the paper. Therefore, we suggest setting $k = 30$ for general use. Regarding $\sigma^2$, we suggest using $\sigma^2 = 1.0$ for general applications as its variation has a minimal impact on the results.

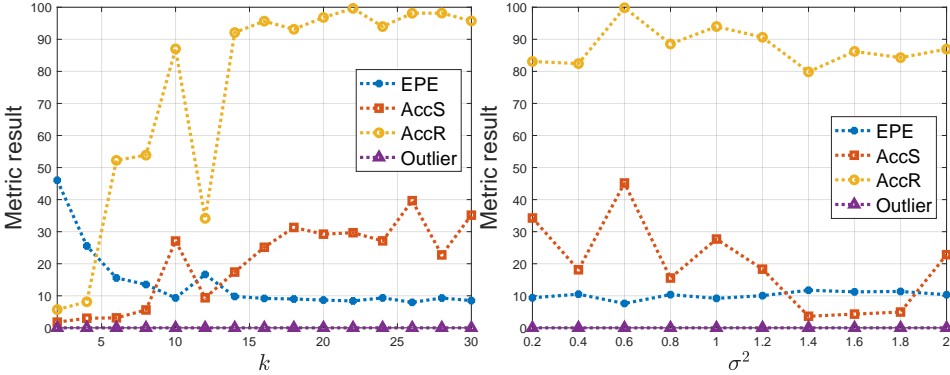

Figure 8: Influences of $k$ and $\sigma^2$ on non-rigid registration under occlusion disturbance scenarios.

Table 4: Ablation study of the regularization effect of the activation functions on the occluded Open-CAS liver dataset. ↑ means larger values are better while ↓ means smaller values are better.

| Metric | Liver 1 | | | | Liver 2 | | | | Liver 3 | | | |
|---|---|---|---|---|---|---|---|---|---|---|---|---|
| Method | EPE ↓ | AccS ↑ | AccR ↑ | Outlier ↓ | EPE ↓ | AccS ↑ | AccR ↑ | Outlier ↓ | EPE ↓ | AccS ↑ | AccR ↑ | Outlier ↓ |
| ReLU+CD | 47.994 | 9.106 | 20.466 | 17.957 | 39.171 | 9.684 | 23.321 | 8.533 | 27.494 | 28.766 | 44.443 | 11.272 |
| ReLU+MCC | 31.582 | 28.446 | 40.614 | 4.378 | 12.012 | 47.258 | 69.975 | 0.000 | 17.245 | 32.841 | 51.744 | 1.326 |
| ReLU+MCC+LLR | 14.690 | 34.144 | 57.860 | 0.000 | 6.984 | 66.543 | 89.956 | 0.000 | 14.306 | 36.670 | 52.982 | 0.000 |

Table 5: Ablation study of the intrinsic regularization effect of the activation functions on the complete Open-CAS liver dataset. ↑ means larger values are better while ↓ means smaller values are better.

| Method | Liver 1→Liver 2 | Liver 2→Liver 3 | Liver 3→Liver 1 |
|---|---|---|---|
| ReLU+CD | 4.685 | 5.550 | 6.787 |
| SIREN+CD | 3.912 | 5.063 | 6.183 |

## E    ANALYSIS OF THE REGULARIZATION

Fig. 5 reveals that regularizing the deformation field of occluded regions is mainly from LLR rather than the network. This is because a network alone is insufficient to handle occlusions effectively, often resulting in significant deviations. We conduct additional ablation studies on the OpenCAS dataset to investigate the role of activation functions (*i.e.*, ReLU and SIREN) with the LLR for regularization. The results presented in Tab. 4 reveal that the regularization of deformations in occluded areas is primarily attributed to LLR rather than the activation function, as ReLU+CD and ReLU+MCC still generate significant deformation errors, especially for both the EPE and Outlier metrics.

Moreover, to investigate the intrinsic regularization capabilities of activation functions, we conduct an additional experiment by applying recursive deformations to complete shapes without occlusion. The EPE metric detailed in Tab. 5 demonstrates that ReLU and SIREN exhibit comparable performance on complete shapes. This finding further substantiates our conclusion that while activation functions do possess a regularization effect, they may fall short when addressing the complexities of challenging occlusions (*i.e.*, the motivation and core contributions of this work). In such cases, LLR proves to be particularly beneficial.

## F    DIFFERENCES AND CONNECTIONS WITH ROBUST FUNCTIONS

Moreover, we conduct an in-depth analysis comparing correntropy with robust functions.

**Differences.**    Unlike common robust functions, correntropy is initially proposed in information-theoretic learning to handle *nonzero mean and non-Gaussian noise* (*e.g.*, the occlusion part can be seen as a certain type of non-Gaussian noise), which is related to the Renyi's quadratic entropy. Besides, MCC is a *local measure* that provides a *probabilistic meaning* of maximizing the error probability density at the origin according to the information potential.

**Connections.**    We derive the relationship between MCC and robust functions. By setting $\rho(e) = \left(1 - \exp\left(-e^2/2\sigma^2\right)\right)/\sqrt{2\pi}\sigma$, we can prove that $\rho(e)$ is an influence function satisfying all the criteria of a robust function, *i.e.*,

$$\rho(e) \geq 0, \rho(0) = 0, \rho(e) = \rho(-e),$$
$$\rho(e_i) \geq \rho(e_j) \text{ for } |e_i| > |e_j|. \tag{17}$$

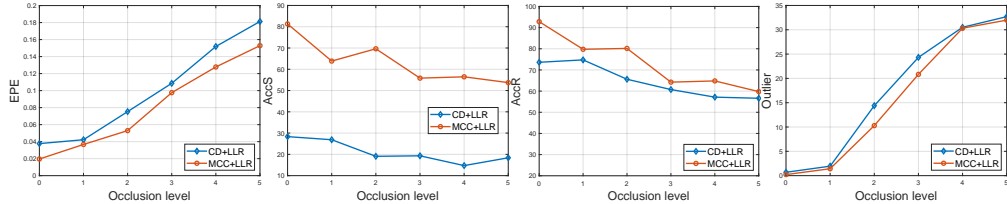

Figure 9: Ablation study of the MCC and the CD metrics.

Moreover,

$$\min_{\theta} \sum_{i=1}^{N} \rho(e_i) \Leftrightarrow \max_{\theta} \sum_{i=1}^{N} \exp\left(-e_i^2/2\sigma^2\right)/\sqrt{2\pi}\sigma \tag{18}$$

with the corresponding weight function of $\rho(e)$ as $w(e) = \exp\left(-e^2/2\sigma^2\right)/\sqrt{2\pi}\sigma^3$. For comparison, the Bi-square's weight function is

$$w_{\mathrm{Bi}}(e) = \begin{cases} (1-(e/h)^2)^2, & |e| \leq h \\ 0, & |e| \geq h \end{cases} \tag{19}$$

where $h$ is the tuning threshold. Therefore, the nonzero part of $w_{\mathrm{Bi}}(e)$ is equivalent to (with a constant) the square of the first-order Taylor expansion of $w(e)$. MCC is analogous to a robust function but with the specific influence function $\rho(e)$. However, unlike the Huber or Bi-square function, correntropy does not need a predefined threshold such as $h$ and the kernel size entirely governs the properties. Moreover, our derivation between correntropy and robust functions offers a practical way to select an appropriate threshold for robust functions or determining the kernel size for correntropy.

## G  ABLATION STUDY OF MCC AND THE CHAMFER DISTANCE

Additionally, we conduct additional ablation experiments to replace MCC with CD on a series of models used in Fig. 5, testing them with progressively increasing levels of occlusion (0-5). The quantitative comparison results in Fig. 9 demonstrates that MCC consistently achieves higher-quality deformation results, particularly with significantly high AccS and AccR metrics. Moreover, we provide a detailed comparison of the time consumption of both the Chamfer Distance and MCC in Tab. 6, measured in seconds. As observed, MCC also achieves significantly high efficiency with the computation time around 1 millisecond for a dataset comprising around $10^4$ points.

Table 6: Comparison of time for MCC and CD metrics in seconds (s).

| Cases | 0 | 1 | 2 | 3 | 4 | 5 | Ave. |
|---|---|---|---|---|---|---|---|
| CD ($\times 10^{-3}$s) | 0.696 | 0.676 | 0.685 | 0.687 | 0.697 | 0.690 | 0.689 |
| MCC ($\times 10^{-3}$s) | 1.332 | 1.327 | 1.466 | 1.323 | 1.285 | 1.316 | 1.342 |

## H  FURTHER COMPARISON WITH OTHER CD VARIANTS

To further demonstrate the advantages and versatility of the MCC metric, we conduct additional experiments to evaluate MCC against the most recent or state-of-the-art *Hyperbolic Chamfer Distance* (HCD) in point cloud completion (Lin et al., 2023). We use the author's source code to implement the HCD metric. Particularly, we adopt three distinct tests by adjusting the coefficient $\alpha$ within the HCD framework to achieve comprehensive results. The quantitative results reported in Tab. 7 demonstrate that our proposed method consistently achieves higher-quality deformations. Moreover, MCC shows potential for further exploration and application in general point analysis tasks, including point cloud completion.

Table 7: Quantitative comparisons between the MCC and the HCD metrics on the occluded Open-CAS liver dataset. ↑ means larger values are better while ↓ means smaller values are better. **Bond** fonts indicate the top performer.

| Metric | Liver 1 | | | | Liver 2 | | | | Liver 3 | | | |
|---|---|---|---|---|---|---|---|---|---|---|---|---|
| Method | EPE ↓ | AccS ↑ | AccR ↑ | Outlier ↓ | EPE ↓ | AccS ↑ | AccR ↑ | Outlier ↓ | EPE ↓ | AccS ↑ | AccR ↑ | Outlier ↓ |
| HCD with $\alpha = 1.0$ | 28.946 | 15.533 | 33.077 | 0.054 | 29.164 | 15.455 | 32.369 | 0.169 | 29.384 | 15.042 | 33.103 | 0.216 |
| HCD with $\alpha = 0.5$ | 27.116 | 12.727 | 33.527 | 0.000 | 26.451 | 11.944 | 32.145 | 0.000 | 26.815 | 12.826 | 32.750 | 0.000 |
| HCD with $\alpha = 0.3$ | 23.535 | 10.414 | 33.338 | 0.000 | 23.074 | 9.730 | 32.881 | 0.000 | 24.291 | 10.872 | 33.744 | 0.000 |
| MCC (Ours) | 8.662 | 29.228 | 96.813 | 0.000 | 5.687 | 75.193 | 97.184 | 0.000 | 12.112 | 42.372 | 56.564 | 0.000 |

## I HOW TO INCORPORATE THE GEOMETRIC DESCRIPTOR

Given the two matching sets $\mathcal{S}(\mathbf{X})$ and $\mathcal{S}(\mathbf{Y})$ extracted by the pre-trained geometric descriptor Lepard, with $|\mathcal{S}(\mathbf{X})| = |\mathcal{S}(\mathbf{Y})| < \min(|\mathbf{X}|, |\mathbf{Y}|)$, our optimization function is formally defined as

$$\mathbf{\Theta}^* = \arg\min_{\mathbf{\Theta}} \mathcal{F}(\mathbf{\Theta}) + \beta \mathcal{M}(\mathbf{\Theta}), \tag{20}$$

where $\mathcal{F}(\mathbf{\Theta})$ and $\mathcal{M}(\mathbf{\Theta})$ denote the deformation loss and the pointwise matching loss, separately.

## J LIMITATIONS AND FUTURE WORK

While the proposed registration method demonstrates impressive performance on various occlusion disturbances, it is worth noting that the problem itself is highly challenging and severely ill-posed. When the occlusion region is significantly large, the method may yield unsatisfactory results (Fig. 18). However, integrating shape analysis theory, such as the *geometric symmetry* of models, could potentially mitigate this limitation.

Besides, the computational complexity of the locally linear reconstruction step is relatively high due to the matrix inverse operation. One possible approach is to explore *low-rank approximations* of the Gram matrix to further speed up the deformation process.

Another future work is to further speed up the computation of the MCC metric. As demonstrated in (Li et al., 2023), a distance transform is utilized to expedite the computation of the Chamfer distance, and in principle, this manner can be extended to the MCC-induced metric as our proposed method also incorporates pairwise computation. However, as the authors noted, we must carefully consider the trade-off between discretization error, grid resolution, memory consumption, and estimation accuracy.

## K MORE QUANTITATIVE AND QUALITATIVE RESULTS

We provide additional results in the following to show the superiority of the proposed method. Specifically,

1. Fig. 10 to Fig. 11 showcase the test results on the occluded TOSCA dataset (Bronstein et al., 2008), where various occlusion scenarios are present.

2. Fig. 12 demonstrates the registration results of our proposed method by inversely deforming the occluded geometry to the complete one. It can be seen that our method achieves highly accurate registrations, preserves the original topology of the organs (including their distinct parts), and avoids the generation of physically infeasible parts.

3. Fig. 13 presents qualitative results on the 4DMatch or 4DLoMatch datasets (Li & Harada, 2022a), where our method performs impressively despite the existence of challenging scenarios such as large deformation, low overlap, and occlusion.

4. Fig. 14 showcases additional shape completion results by deforming the source model (Yang et al., 2014) to the shapes with occluded holes. Thanks to the highly ac-

curate registration results, our method provides a viable solution to this challenging mesh hole filling problem.

5. Fig. 15 further reports the mesh hole filling results where the source or template models also contain holes. The source models are selected from Fig. 7 and Fig. 14, with the middle model in each case serving as the mean shape. It is observed that, due to the superior non-rigid deformation capabilities of our method, the majority of the holes in the original shapes have been effectively filled. Moreover, we present the boolean shapes, which are obtained from the union operation between the deformed surfaces and their corresponding target models (that is, Shape 1 ∪ Result 1 and Shape 2 ∪ Result 2). This boolean operation not only further completes the holes but also demonstrates the seamless integration of the deformed shapes with their target models.

6. Fig. 16 to Fig. 18 report the registration results of our method on real-world RGB-D point clouds under occlusion disturbance.

7. Fig. 19 and Fig. 20 showcase qualitative registration results under significant disturbances caused by noise and outliers, respectively. As demonstrated, our method continues to achieve highly accurate deformations, attributable to the adaptive correntropy function.

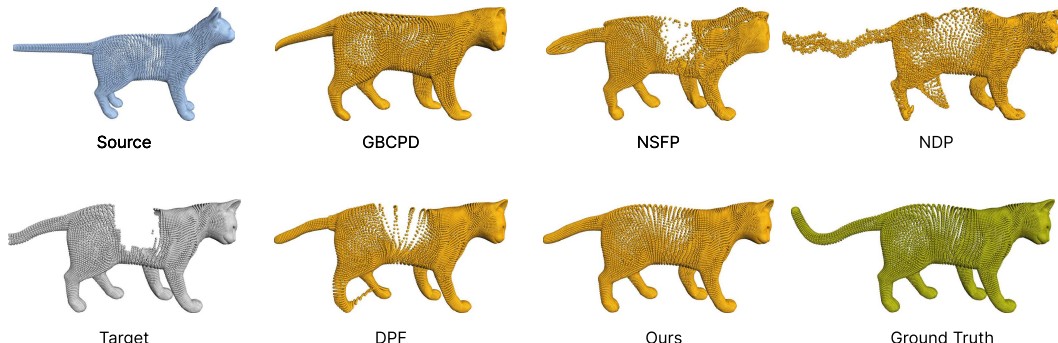

Figure 10: Qualitative comparisons on the occluded cat dataset, where the **body** and **tail** of the cat are occluded.

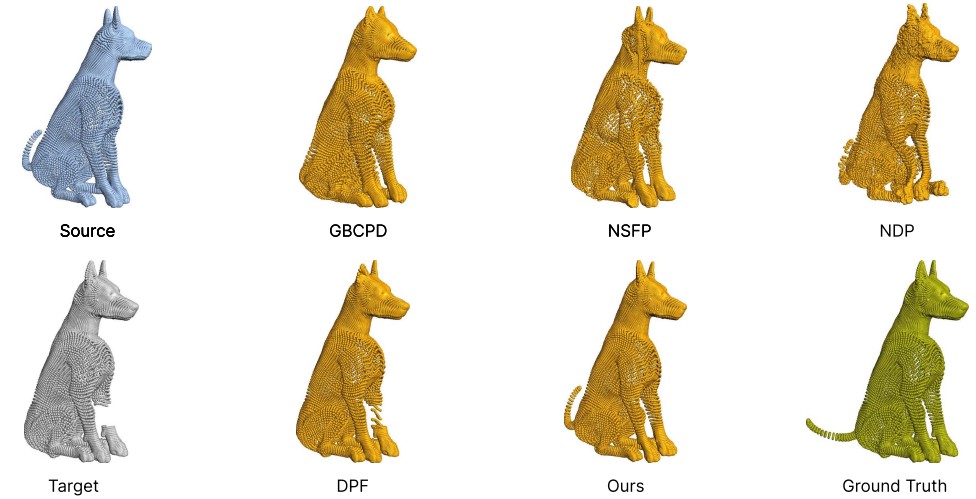

Figure 11: Qualitative comparisons on the occluded dog dataset, where the **left front leg** and the **tail** of the dog are occluded.

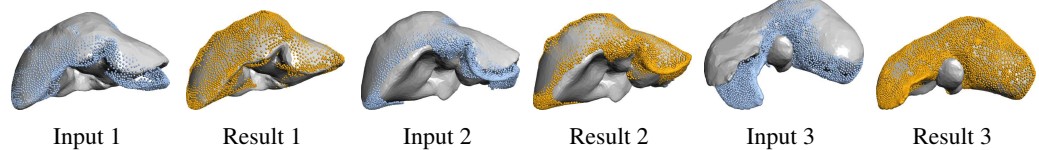

| Input 1 | Result 1 | Input 2 | Result 2 | Input 3 | Result 3 |

Figure 12: Application of the proposed method to register occluded livers to the complete ones. As observed, our method not only delivers highly-accurate registration results but also preserves the original topology of the organs, including their distinct parts, while avoiding the generation of additional physically infeasible parts.

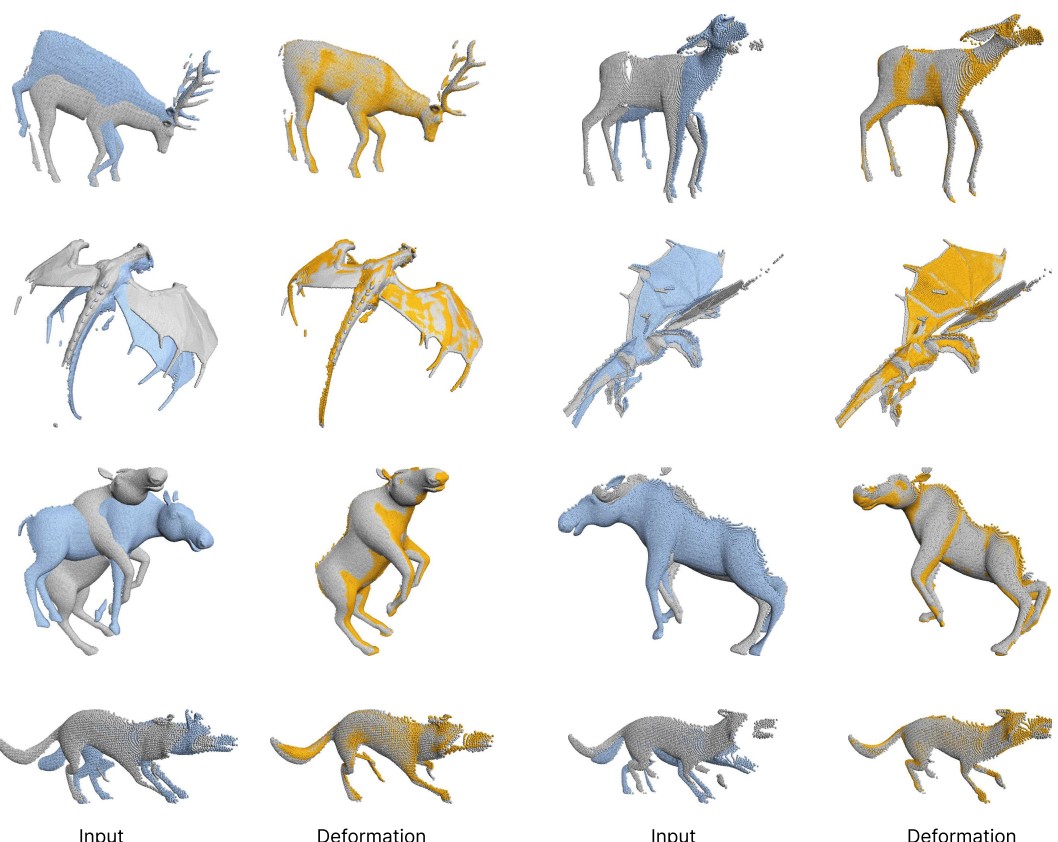

| Input | Deformation | Input | Deformation |

Figure 13: Non-rigid registration on point clouds from the challenging 4DMatch or 4DLoMatch datasets, which exhibit **significant deformations**. The blue and gray point clouds represent the source and target models, separately, while the yellow color indicates our registrations. Our method still yields satisfactory results even in the presence of substantial deformations.

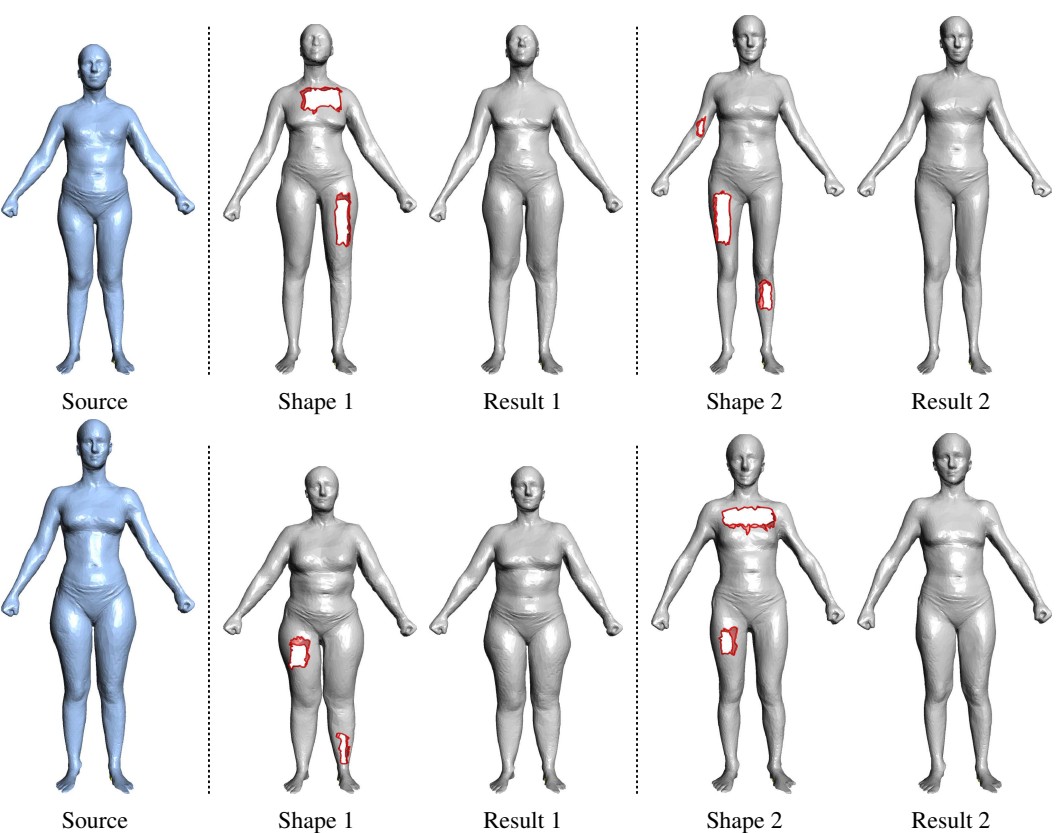

Figure 14: Application of the proposed method to mesh hole filling by deforming the source model to the target shape.

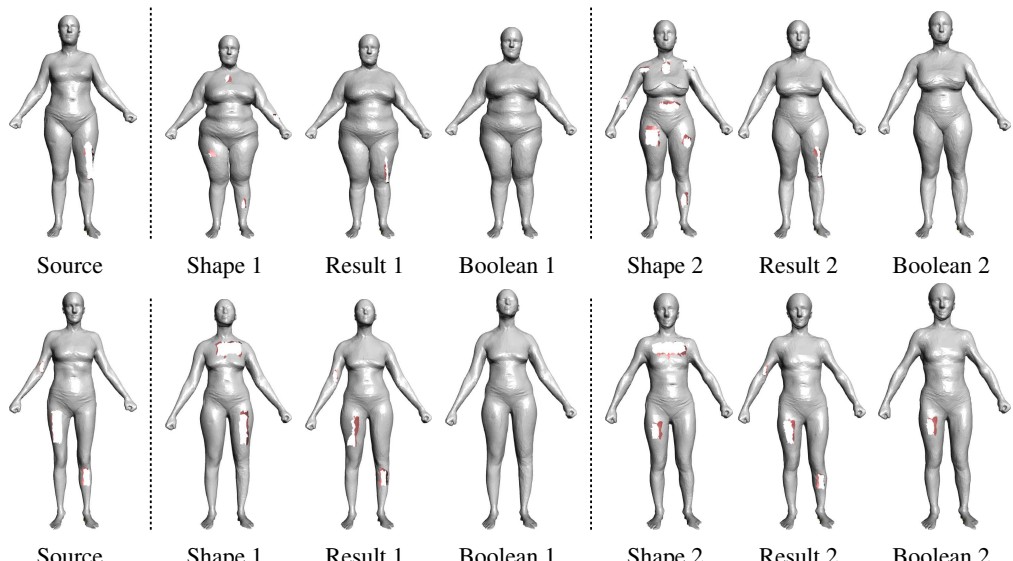

| Source | Shape 1 | Result 1 | Boolean 1 | Shape 2 | Result 2 | Boolean 2 |

Figure 15: Application of the proposed non-rigid registration method to mesh hole filling. This figure illustrates the process of deforming a source model containing holes to align with other models that also have holes. "Result 1" and "Result 2" represent the deformed shapes by our non-rigid registration method, which consistently delivers high-quality deformations regardless of the holes. "Boolean 1" and "Boolean 2" are obtained by taking the union operation of the deformed surfaces with their respective targets (*i.e.*, Shape 1 ∪ Result 1 and Shape 2 ∪ Result 2), highlighting the seamless integration of the deformed shapes with their target models.

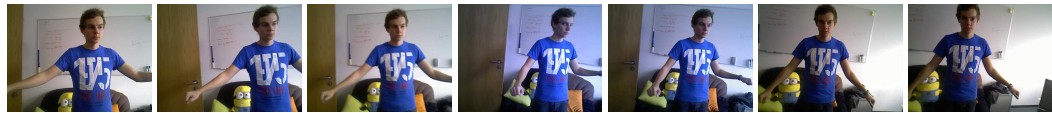

Figure 16: RGB sequence images with occlusion used in generating real RGB-D point clouds.

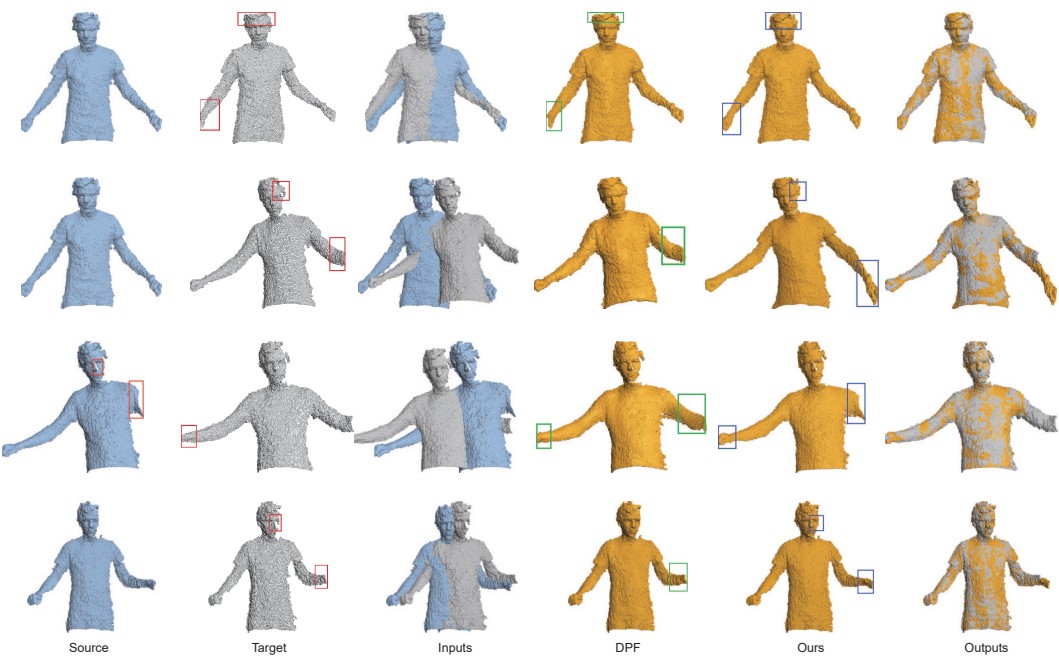

Source      Target      Inputs      DPF      Ours      Outputs

Figure 17: Non-rigid registration on RGB-D point clouds with occlusion indicated by the red box. As observed, DPF suffers from physically implausible issues such as **collapse or pull** of the body parts as shown in the green box, instead, our method not only delivers successful registration, but also preserves the **physical rationality** of the human body, such as the hands and arms in the blue box.

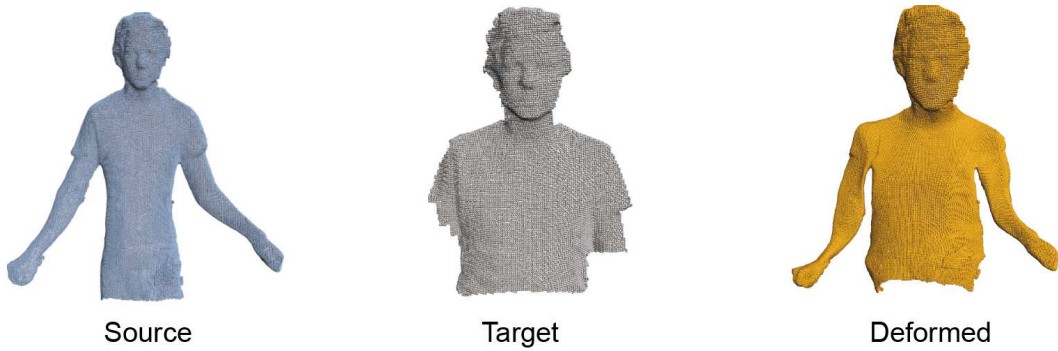

Source               Target              Deformed

Figure 18: Failure cases due to significantly large occlusion (*i.e.*, the occlusion of **both human arms and body**). Nonetheless, our method preserves high-quality deformation in the overlapping regions and still strives to deliver **physically plausible deformation** for occluded areas to the best extent possible.

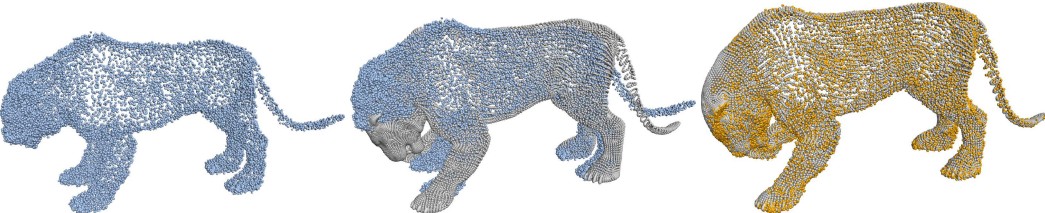

Figure 19: Non-rigid point cloud registration under **heavy noise disturbances** (intensity=0.6%). Our method still delivers satisfactory registration thanks to the utilized adaptive correntropy function.

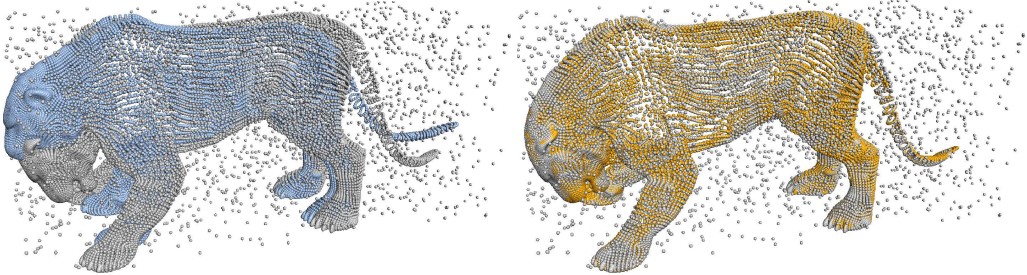

Figure 20: Non-rigid point cloud registration under **heavy outlier disturbances** (#outliers=2,350). Our method exhibits substantial robustness against outliers, attributed to the introduction of the adaptive correntropy function.

