# OpenReview forum: "Occlusion-aware Non-Rigid Point Cloud Registration via Unsupervised Neural Deformation Correntropy"
_ICLR.cc/2025/Conference — ICLR 2025 Poster_

### Official Review · Reviewer_Rxao · 2024-10-29

**Soundness:** 3
**Presentation:** 2
**Contribution:** 2
**Rating:** 6
**Confidence:** 4

**Summary:**

In this paper, the authors propose an occlusion-aware non-rigid point cloud registration method based on cross correntropy between deformed source shape $T(Y)$ and occluded target shape $X$.  The paper argues that the main failure reason of previous non-rigid point cloud registration methods is that most of them are based on the standard Chamfer Distance (CD), which can lead to collapsed or physically implausible results. To overcome the limitation, the paper demonstrates theoretical analysis between maximum correntropy criterion (MCC) and CD and concludes that the standard CD is a special case of MCC and MCC is more robust in occluded regions. Furthermore, the paper proposes a regularization based on locally linear reconstruction (LLR) inspired by local linear embedding (LLE) to regularize the deformation smoothness of unmatched points and experimentally demonstrate the superior performance of LLR compared to as-isometric-as-possible (AIAP). In the experiment part, the paper conducts extensive experiments under different settings (e.g. medical data, human, animal, different level of occlusions) and demonstrates the superior performance of the proposed method compared to prior works, including both axiomatic and learning-based methods.

**Strengths:**

1. The paper conducts exhaustive experiments in different scenarios ranging from medical data to animal and humans and demonstrates superior performance of the proposed method compared to both axiomatic and learning-based method.
2. The idea of using maximum correntropy criterion (MCC) and local linear reconstruction (LLR) for non-rigid point cloud registration is novel and technically sound. In the ablation study, the paper also demonstrates the superior performance of LLR in comparison to as-isometric-as-possible (AIAP) regularization, which is commonly used in previous methods.

**Weaknesses:**

1. In the experiment part, the paper only demonstrates rather synthetic point cloud datasets in the context of occlusion ratio as well as point cloud sampling density. When it comes to more challenging dataset like 4DMatch and 4DLoMatch with less overlap and large motions. The proposed method needs to use the pre-trained geometric feature descriptor Lepard. Meanwhile, it would be better to shortly describe how to incorporate the pre-trained feature descriptor into the proposed method.
2. The motivation of using neural implicit representation is not well illustrated. To my understanding, the proposed regularization can also be directly used to optimize the deformation field defined on each point in the point cloud.
3. It would be better to conduct an ablation study, which replaces MCC with standard Chamfer Distance to better demonstrate the effectiveness of MCC.
4. In the literature of point cloud completion, some works also propose variants of CD to address the problem of occlusion, it would be better to also compare MCC with their proposed variants, e.g.
T. Wu, et al.: Density-aware Chamfer Distance as a Comprehensive Metric for Point Cloud Completion (NeurIPS 2021)
F. Lin, et al.: Hyperbolic chamfer distance for point cloud completion (ICCV 2023)

**Questions:**

1. In Eq.3, the definition of $\tilde{y}_{i}$ should be  $T(Y)$ rather than $Y$?
2. More details of Lemma 1 should be provided, since it is the crucial difference between MCC and standard CD.

---

> ### Author Response · Authors · 2024-11-18
> **Part 1: Detailed Answer and Discussions**
>
> Thank you very much for your encouraging comments and constructive suggestions. We have conducted additional experiments and give a detailed response in the following. Due to space limit, we first address the weakness concerns of our method, followed by responses to questions in the subsequent Part 2. We sincerely hope that we have addressed your concerns.
>
> #  Re: Weaknesses
>
> ---
>
> - **Incorporation of Geometric Descriptor and how to Use it:** Thanks. The core contribution of this work is the optimization of a **non-rigid deformation field** designed to align two shapes, especially under conditions of **occlusion**. This is somewhat different from geometric matching methods, which primarily focus on establishing **correspondences** between shapes. Deformation methods are often considered a subsequent step after the matching process to deal with large motions, as has been adapted in many previous approaches. However, our method is more effective for shapes with occlusion. We also demonstrate its versatility by  integrating it with state-of-the-art correspondence methods.
>
>   In our experiments, following [1], we employ a pre-trained descriptor for initial landmark point matching. Specifically, given the two matching sets  $\mathcal{S}(\mathbf{X})$ and $\mathcal{S}(\mathbf{Y})$ with $|\mathcal{S}(\mathbf{X})|=|\mathcal{S}(\mathbf{Y})|<\min(|\mathbf{X}|,|\mathbf{Y}|)$, our optimization function is formally defined as
>   $$\mathbf{\Theta}^*=\mathop{\arg\min}\limits_{\mathbf{\Theta}}\mathcal{F}(\mathbf{\Theta})+\beta\mathcal{M}(\mathbf{\Theta})$$
>   where $\mathcal{F}(\mathbf{\Theta})$ and $\mathcal{M}(\mathbf{\Theta})$ denote the deformation loss and the pointwise matching loss, separately. We have added these explanations in Sec. I of the Appendix to make it more understandable.
>
> - **Motivation for Utilizing Implicit Neural Fields (INFs):** We appreciate the inquiry. There are several reasons for employing INFs in our work:
>
>    (1). INFs leverage the universal approximation capability of neural networks to effectively describe deformation fields. This **eliminates the need to manual and explicit definition** of complex deformations, such as the traditional spline and kernel function methods [1].
>
>    (2). The adoption of INFs enables **automatic differentiation**, which brings powerful optimization efficacy.
>
>    (3). INFs also offer the advantage of **runtime optimization**, allowing for an **unsupervised** approach that ensures robust generalization.
>
>    Yes, as LLR enables reasonable deformations even for unmatched parts, which in turn naturally ensures the optimization and smoothness of the deformation field for each point.
>
>
> - **Ablation Study of MCC and CD:** Thanks for this suggestion. We have conducted additional ablation studies to demonstrate the superiority of the MCC metric against CD and added these experiments in the revised manuscript. Please refer to the first response to Reviewer FdAL for details.
>
>
> - **Further Comparison with the Suggested CD Variants:** Thanks. We have conducted further experiments to evaluate MCC against the most recently or state-of-the-art Hyperbolic Chamfer Distance (HCD) in point cloud completion [2]. We use the author's source code to implement the HCD metric. Particularly, we adopt three distinct tests by adjusting the coefficient $\alpha$ within the HCD framework to achieve comprehensive results. The quantitative results reported in the following table demonstrate that our proposed method consistently achieves higher-quality deformations. Moreover, MCC shows potential for further exploration and application in general point analysis tasks, including point cloud completion.
> - |Method   |EPE↓|AccS↑| AccR↑|Outlier↓|EPE↓|AccS↑|AccR↑|Outlier↓| EPE↓|AccS↑|AccR↑|Outlier↓|
> |-|--|-|-|-|-|-|-|-|-|-|-|-|
> |HCD with $\alpha=1.0$ |28.946| 15.533| 33.077|  0.054|[29.164| 15.455| 32.369|  0.169| 29.384| 15.042| 33.103| 0.216|
> |HCD with $\alpha=0.5$|27.116| 12.727|33.527| 0. 000  |26.451| 11.944| 32.145|0.000|26.815| 12.826| 32.750|  0. 000|
> |HCD with $\alpha=0.3$|23.535|10.414| 33.338|0.000|23.074|9.730|32.881|0. 000      |24.291|10.872|33.744|0. 000
> |MCC (Ours) |**8.662**|**29.228**|**96.813**|**0.000**|**5.687**|**75.193**|**97.184**|**0.000**|**12.112**|**42.372**|**56.564**|**0.000**
>
>   We have added these experiments and analyses in Sec. H of the Appendix of the revised manuscript.
>
> # References
> ---
> [1] Yang et al., Non-Rigid Point Cloud Registration with Neural Deformation Pyramid, NeurIPS, 2022.
>
> [2] Myronenko et al., Point Set Registration: Coherent Point Drift, IEEE Transactions on Pattern Analysis and Machine Intelligence, 2010.
>
> [3] F. Lin, et al., Hyperbolic Chamfer Distance for Point Cloud Completion, ICCV, 2023.

---

> ### Author Response · Authors · 2024-11-19
> **Part 2: Further Answer and Discussions**
>
> We give a detailed answer to other comments and sincerely hope that we have addressed your concerns.
>
>
> # Re: Questions
>
> ---
>
> - **Update $\mathcal{T}(\mathbf{Y})$:** Thanks for this detailed comment. We have revised it in the update manuscript.
>
> - **Adding More Details of Lemma 1:** Thanks. We can rewrite the correntropy as
> $$
> V_\sigma(X, Y)=1/(\sqrt{2 \pi} \sigma) \sum_{n=0}^{\infty}\left((-1)^n\right) /\left(2^n n!\right) \mathbb{E}\left[\left((X-Y)^{2 n}\right) /\left(\sigma^{2 n}\right)\right]$$
> which involves all the even moments of the random variable $X-Y$. When two points are in close proximity, the second-order moment tends to be dominant, causing correntropy to approach correlation, as the higher-order moments decay more rapidly. As the points become more distant, as shown in [1], correntropy transitions from behaving like an $\ell_1$ norm to resembling an $\ell_0$ norm. This transition further illustrates the inherent robustness of the MCC metric. We have addd these details and explanations in Sec. C of the Appendix of the update manuscript.
>
> # References
> ---
>
> [1] Liu et al., Correntropy: Properties and Applications in Non-Gaussian Signal Processing, IEEE Transactions on Signal Peocessing, 2007.

---

> > ### Comment · Reviewer_Rxao · 2024-11-23
> >
> > Thanks for your detailed response, which resolves all of my questions.

---

> > > ### Author Response · Authors · 2024-11-24
> > > **Thank You Vey Much for Your Response**
> > >
> > > Dear Reviewer Rxao,
> > >
> > > Thank you very much for the follow-up!
> > >
> > > We appreciate the discussion, your feedback, and your suggestions. Many thanks for your time and effort.
> > >
> > > Best and sincere wishes,
> > >
> > > The authors

---

### Official Review · Reviewer_C5yq · 2024-10-29

**Soundness:** 3
**Presentation:** 3
**Contribution:** 3
**Rating:** 6
**Confidence:** 4

**Summary:**

The paper proposes an unsupervised method for deformable point cloud registration, aiming to handle the case where parts of the point cloud are missing due to occlusions. Prior works have mainly used the Chamfer Distance to measure the similarity between the registered point clouds, which does not account for occlusion. In contrast, the proposed method utilizes the correntropy-based metric. This metric includes a decaying kernel function that reduces the influence of the occluded region on the deformation result and enables the registration of point clouds with occlusion. To accommodate the deformation of the occluded parts, the authors include a linear reconstruction regularization term to preserve the original point local structure in the deformed point cloud. The point deformation field is parametrized as a coordinate-based neural network, where the network's parameters are optimized subject to the correntropy and regularization term. The method is applied to several point cloud benchmarks with occlusion, as well as for shape interpolation and completion applications, and demonstrates favorable performance.

**Strengths:**

The paper is well-written and easy to understand. The method is technically sound, can be applied to various applications, and improves over the compared alternatives. That said, important evaluations and comparisons are missing.

**Weaknesses:**

The method is evaluated quantitatively on the Open-CAS, 4DMatch, and 4DLoMatch, and several shapes from the TOSCA dataset. Still, a major dataset in the non-rigid deformation literature is SHREC’19 [1]. The shape completion application hints that the method can handle human body non-rigid matching and an evaluation on this dataset can further demonstrate the method's utility. Additionally, the evaluation on partial animal shapes should be done with the common SHREC'16 benchmark [2] instead of several selected shapes from TOSCA.

Comparison:
There are prior works that use an implicit field [3] or self-supervision [4] for non-rigid partial shape registration. Such works should be discussed and compared.

I think the paper passes the acceptance threshold, though adding the missing evaluations and comparisons can strengthen the submission much further.

[1] Melzi, S., Marin, R., Rodol`a E., Castellani U., Ren J., Poulenard A., Wonka P., Ovsjanikov M. SHREC 2019: Matching humans with different connectivity. In Eurographics Workshop on 3D Object Retrieval. vol. 7 (2019)
[2] Cosmo L., Rodol`a E., Bronstein M M., Torsello A., Cremers D., Sahillioglu Y. SHREC '16: Partial matching of deformable shapes. In Proc. 3DOR, 2(9):12 (2016).
[3] Cao D, Bernard F. Self-supervised learning for multimodal non-rigid 3d shape matching. In Proceedings of the IEEE/CVF Conference on Computer Vision and Pattern Recognition (2023).
[4] Sundararaman, R., Pai, G., Ovsjanikov, M. Implicit field supervision for robust non-rigid shape matching. In the European Conference on Computer Vision (2022).

**Questions:**

Please address the concerns raised in the "weakness" section.

---

> ### Author Response · Authors · 2024-11-20
> **Detailed Answer and Discussions**
>
> Thank you very much for your encouraging comments and constructive suggestions.  We  have conducted additional experiments to test the proposed method and sincerely hope that we have addressed your concerns.
>
> # Re: Weaknesses
> ---
>
> - **Test on the SHREC'19 Dataset:** Thanks. We have performed additional experiments on the SHREC'19 dataset, which includes 44 shapes and a total of 430 evaluation pairs. Following the suggested work, we report the mean geodesic errors in the subsequent table. It is noteworthy that, while our focus in this work is on **optimizing a non-rigid deformation field** with a particular emphasis to ensure physically plausible deformations for occluded parts, rather than on **shape matching**, our method still yields comparatively satisfactory results (albeit slightly inferior to those of specialized matching approaches). The results also demonstrate that our method can indeed provide an appropriate **initialization** which can then be enhanced through subsequent matching optimization processes.
>
>
> - |Methods|SMM[1]|IFMatch[2]|Ours|
> |-|--|-|-|
> |Geodesic error|0.040|0.065|0.113|
>
> - **Test on the SHREC'16 Dataset:** Thanks. We have conducted further experiments on partial shapes from the challenging SHREC’16 dataset, which contains 200 test shapes categorized into CUTS and HOLES types. The mean geodesic errors reported in the following table reveal that methods **specifically tailored for shape matching**, such as SMM, achieve higher accuracy. However, our method is capable of providing a suitable initialization for these challenging cases, even without pretraining.
>
> - |Methods|CUTS|HOLES|
>   |-|-|-|
>   |SMM|0.076|0.159|
>   |Ours|0.123|0.284|
>
>
> - **Differences from and Discussions of the Suggested Methods:** Thanks. There are two core contributions of our work: 1) The **local and adaptive** metric MCC effectively **prevents physically implausible deformations**, such as collapses and tearing, which are common in previous non-rigid registration methods. 2) We employ LLR to further **ensure reasonable deformations of occluded parts**, a problem that previous approaches have not adequately addressed. In contrast to our method, [1] primarily focuses on training a self-supervised network for multimodal shape matching. While it shows promising performance, it does not address the deformations of occluded parts. [2] utilizes an auto-decoder structure to implicitly align two volumes, which requires surface normals for training. Additionally, it employs a bi-directional Chamfer Distance for inferences, a metric that may be susceptible to occlusion. In comparison, our method stands out as an **unsupervised**, **runtime-optimization** approach, which ensures **robust generalization** without the need of labeled data for specific training. We have cited and added discussions of the suggested work in Sec. Related Work of the revised manuscript.
>
> # References
> ---
>
> [1] Cao et al., Self-supervised learning for multimodal non-rigid 3d shape matching, CVPR 2023.
>
> [2] Sundararaman et al., Implicit field supervision for robust non-rigid shape matching, ECCV 2022.

---

> > ### Comment · Reviewer_C5yq · 2024-11-27
> >
> > Thanks. I have no further questions.
> > Given the rest of reviews/answers, I still find it puzzling that LLR has such an impact. Interesting observation.
> > I'll keep my score.

---

> > > ### Author Response · Authors · 2024-11-28
> > > **Thank You Very Much for Your Response**
> > >
> > > Dear Reviewer C5yq,
> > >
> > > Thank you very much for the follow-up!
> > >
> > > We pioneer the use of LLR to ensure physically reasonable deformations for occluded parts, which has not been explored by previous approaches.
> > >
> > > We appreciate the discussion, your feedback, and your suggestions. Many thanks for your time and effort.
> > >
> > > Best and sincere wishes,
> > >
> > > The authors

---

> ### Author Response · Authors · 2024-11-24
> **End of Discussion Approaching**
>
> Dear Reviewer C5yq,
>
> Thank you very much for reviewing our paper. Given that the end of the discussion period is approaching, we would like to ask if you have any further concerns or questions, particularly as a follow-up or update the score to our response?
>
> We appreciate the discussion, your feedback, and your suggestions. Many thanks for your time and effort.
>
> Best and sincere wishes,
>
> The authors

---

### Official Review · Reviewer_FdAL · 2024-10-30

**Soundness:** 3
**Presentation:** 3
**Contribution:** 3
**Rating:** 6
**Confidence:** 3

**Summary:**

This paper proposed a novel method for non-rigid registration between two point clouds under partial occlusion. The core idea of the proposed method is to use the Maximum Correntropy Criterion (MCC) to measure the similarity between two point clouds. By the combination of MCC and implicit neural representations, efficient unsupervised point cloud registration algorithm can be developed. Furthermore, a local linear reconstruction (LLR) formulation has been utilized to regularize the deformation of occluded region of point cloud. Experiment on the Open-CAS liver and several self-built dataset shows that the proposed method achieves impressive result under occlusion and large deformation.

**Strengths:**

1. Although the MCC is an existing metric, but its usage in the framework of implicit neural deformation optimization is simple, elegant and effective
2. The experiment result is impressive, the proposed method consistently outperforms competing approaches by large margin on Open-CAS liver dataset
3. The proposed LLR has demonstrated its effectiveness in ablation study, especially in high level of occlusion
4. The presentation is clear and easy to follow, and the result is reproducible with source code

**Weaknesses:**

1. There is no ablation study on the use of MCC and Chamfer distance for registration. As Proposition 1 says that the Chamfer distance is a special case of MCC induced metric, it would be good to see how the generalization of the MCC induced metric works on real-world point cloud registration, for example, with different level of occlusion, deformation, or even registration in same-class level, not same-instance level. It would also be good to see the comparison on computational efficiency
2. Table 4 is in appendix, not in the main text
3. The shape completion task in section 5.7 requires a complete source mesh model as template, which may not be available in practical scenarios. It would be good to see whether a 'mean' shape mesh model could serve as the template and achieve good mesh hole filling result.

**Questions:**

1. In Fig.6 shape interpolation experiment, what the timestamp t means? is it some mixture ratio between two shapes? it would be good to briefly describe the settings for shape interpolation

---

> ### Author Response · Authors · 2024-11-14
> **Ablation Study and Mean Shape Hole Filling**
>
> Thank you very  much for your encouraging comments and valuable suggestions. We have conducted additional experiments and give a detailed response in the following. We sincerely hope that we have addressed your concerns.
>
> # Re: Weaknesses
>
> ---
>
> - **Ablation Study of MCC and the Chamfer Distance:** Thanks for this constructive suggestion. As suggested, we have conducted additional ablation experiments to replace MCC with CD on a series of models used in Fig. 5, testing them with progressively increasing levels of occlusion (0-5). The new added Fig. 9 in the Appendix demonstrates that MCC consistently achieves higher-quality deformation results, particularly with significantly high AccS and AccR metrics. Moreover, we provide a detailed comparison of the time consumption of both the Chamfer Distance and MCC in the subsequent table, measured in **seconds**. As observed, MCC also achieves significantly high efficiency  with the computation time around **1 millisecond** for a dataset comprising around $10^4$ points. We have added these time tests in Sec. G of the Appendix in the updated manuscript.
>
> - |Cases   |0|1|2|3|4|5|Ave.
> |-|-|-|-|-|-|-|-|
> |CD time $(\times 10^{-3}$s)|0.696|0.676|0.685|0.687|0.697|0.6900|0.689|
> |MCC time $(\times 10^{-3}$s)|1.332|1.327|1.466|1.323|1.285|1.316|1.342|
>
> - **Generalization Capability:** Thanks. As an **unsupervised** and **runtime optimization-based** neural deformation pipeline, our method does not require data annotations. It reasons non-rigid deformations from a case-by-case geometric perspective, hence guaranteeing a high generalization to unknown categories. Moreover, indeed we have demonstrated the generalization capabilities of MCC across a variety of real-world point cloud scenarios, such as the human registration on occluded depth views (where different levels of occlusion are present), as detailed in Sec. 5.4, and shape completion applications (i.e., same-class level, not same-instance level), covered in Sect. 5.7. In these tests, the shapes are real data sets captured using scanners, thereby further validating the robust generalization of our method to real-world scenarios.
>
> - **Place Table 4 in the Main Text:** Thank you for this careful comment. We have placed Table 4 in the main text to enhance clarity  in the updated manuscript.
>
> - **Mesh hole filling with a mean shape:** Thanks. We have conducted additional experiments to apply our proposed method to the challenging task of mesh hole filling, particularly in cases where **the source or template models also contain holes**. The source models are selected from Fig. 7 of the main paper and Fig. 14 of the appendix, with the middle model in each case serving as the mean shape.
>
>   As presented in the added Item 5 of Sec. K and Fig. 15 of the updated Appendix, our method consistently achieves high-quality deformation results, as evidenced by the 'Result 1' and 'Result 2' deformed shapes. It is observed that, due to the superior non-rigid deformation capabilities of our method, the majority of the holes in the original shapes have been effectively filled. Moreover, we present the boolean shapes, which are obtained from the union operation between the deformed surfaces and their corresponding target models (that is, Shape 1 $\cup$ Result 1 and Shape 2 $\cup$ Result 2). This boolean operation not only further completes the holes but also demonstrates the seamless integration of the deformed shapes with their target models.
>
> # Re: Questions
>
> ---
>
> - **Explanation of the Timestamp $t$**: Thanks. The timestamp $t\in[0, 1]$ represents the deformation process from the source  shape to the target one. For instance, the interpolation shape with $t=0.5$ is represented by $\mathbf{Y}_{t=0.5}=\mathbf{Y}+t*\nu(\mathbf{Y})=\mathbf{Y}+0.5*\nu(\mathbf{Y})$ with $\nu$  denoting the learned displacement field. We have added these explanations in the revised manuscript to make it more understandable.

---

> > ### Comment · Reviewer_FdAL · 2024-11-25
> >
> > Thanks the authors for the additional experiment on ablation study and mesh hole filling, I think all my concerns have been addressed and I will keep the positive rating

---

> > > ### Author Response · Authors · 2024-11-25
> > > **Thank You Vey Much for Your Response**
> > >
> > > Dear Reviewer FdAL,
> > >
> > > Thank you very much for your response and the follow-up.
> > >
> > > We appreciate the discussion, your feedback, and your suggestions. Many thanks for your time and effort.
> > >
> > > Best and sincere wishes,
> > >
> > > The authors

---

> ### Author Response · Authors · 2024-11-24
> **End of Discussion Approaching**
>
> Dear Reviewer FdAL,
>
> Thank you very much for reviewing our paper. Given that the end of the discussion period is approaching, we would like to ask if you have any further concerns or questions, particularly as a follow-up or update the score to our response?
>
> We appreciate the discussion, your feedback, and your suggestions. Many thanks for your time and effort.
>
> Best and sincere wishes,
>
> The authors

---

### Official Review · Reviewer_Jzhm · 2024-11-03

**Soundness:** 3
**Presentation:** 3
**Contribution:** 2
**Rating:** 5
**Confidence:** 5

**Summary:**

Authors proposed a new approach to non-rigid point alignment using implicit deformation networks. Solving non-rigid point alignment using implicit deformation is attractive as it is self-supervised can generalize to well to unknown categories. There is specific focus is on how best to handle occlusion using such networks. The authors propose a maximum correntropy criterion. They argue that it effectively avoids collapsed, tearing, and other physically implausible results – leading to substantially better results than current state of the art.

The authors also demonstrate impressive results on a variety of benchmarks including 4DMatch and OpenCAS. One concern, however, with the approach is that it is essentially a well documented loss function, and well known regularizer applied to the problem of non-rigid point alignment. There is no real explicit handling of occlusion, it is instead handled implicitly through the loss function and regularizer. Also, there needs to be further ablation  into how the network architecture itself regularizes the solution. This seems to have been discarded as a minor detail in Sec 4.3. As it stands, I am on the fence with the paper. The results are good, but the motivation, ablation and new ideas are lacking.

One major concern I have is the proposed objective seems quite similar to classical robust error functions used for decades within robotics and computer vision literature. It would be good if the authors could relate their loss function to these other methods. My initial feeling is that they would almost get identical results, but happy to hear back from the authors about why I am wrong.

Most famously the Huber loss function,

- P. J. Huber, “Robust Estimation of a Location Parameter,” The Annals of Mathematical Statistics, vol. 35, no. 1, pp. 73–101, 1964.

Huber loss functions and their variants have been used for decades in both 3D point and image alignment problems to deal with occlusion and outliers. A great example can be found in:-

- A. W. Fitzgibbon, “Robust registration of 2D and 3D point sets,” Image and Vision Computing, vol. 21, no. 13-14, pp. 1145–1153, 2003.

Subsequent works have evaluated broad families of robust error functions.

One notable example includes:-

- B. Theobald, I. Matthews, and S. Baker. “Evaluating error functions for robust active appearance models,” in IEEE International Conference on Automatic Face and Gesture Recognition, 2006.

**Strengths:**

- The results are impressive compared to current state of the art (especially in the presence of significant occlusion), and the authors do a good job of applying their approach across a number of application domains.
- I especially like use of LLR in the method, using classical work from Roweis and Saul (’00). The Application of the LLR regularization loss is critical to the success of the propose approach. However, it does lead to questions of where does the regularization of the method is stemming (implicitly from the network, all solely through the LLR loss).
- Authors do a good job on the ablation of their method, especially with respect to the number of neighbors and bandwidth (see Appendix C).
- The evaluation and visualization of results are compelling.

**Weaknesses:**

- In general I do have some concerns over novelty. At the end of the day the authors are proposing an existing loss, and regularizer to a new problem. The novelty is really in how well it works for the problem of non-rigid point alignment.
- In Sec 4.3, the authors use a SIREN style implicit neural function (INF) using sinusoid activations. This departs quite heavily from the NSFP work of Li et al. which use ReLU activations. One motivation in NSFP for ReLU is that it leverages the inherent spectral bias of ReLU to find low-frequency (i.e. smooth) solutions. By using SIREN style INFs the authors would kill this property. It seems this would be an additional motivation for the LLR regularization. Some sort of ablation on the role of activation with the LLR would be useful, as they seem to both be regularizing the solution.
- There seems to be a number of mistakes in the Lemma and Definitions, they also seem quite trivial and do not add much to the paper.
- The authors report timing information for their method in Table 2. NSFP is currently considered quite slow for an INR method. Approaches like Li et al. “Fast Neural Scene Flow” (ICCV’23) show a 30 times speedup with almost no loss in performance. It would be useful for the authors to discuss this, and talk about how feasible such a strategy would be to their method.
- See my other concerns in the summary concerning the relation of the author's loss function with classical robust error functions (e.g. Huber loss).

**Questions:**

- In Definition 1, why are the authors referring to bandwidth? It seems the statement is quite generic (covering all kernels that satisfy Mercer’s theorem). Kernels like Gaussian have bandwidth, but that seems much more restrictive.
- Lemma 1 seems wrong. The authors have not defined the kernel, and just said that it needs to satisfy Mercer’s theorem. It is trivial to show that if k(x,y) = ||x – y||, then it would not be L2 close, L1 as they get apart, and L0 far apart. It seems this statement should be dependent upon the type of kernel used? For example, for the specific case of a Gaussian kernel I would believe such a statement.
- Eq. 2 seems to have a mistake. They use k(x – y), but I suspect it should be k(x,y)?

---

> ### Author Response · Authors · 2024-11-14
> **Part 1: Regularization Analysis and Derivation with Robust Functions**
>
> Thank you very much for your insightful comments and valuable suggestions. Due to space limit, we first give a detailed answer to the two key concerns regarding regularization and the relationship to robust functions and present additional answers in the subsequent Part 2. By these clarifications we sincerely hope that we have addressed your concerns.
>
> # Re: Weaknesses
> ---
> - **Analysis of the Regularization:**  Thanks. Indeed, we have reported an ablation study in the paper (Sec. 5.6) to reveal that regularizing the deformation field of occluded regions is mainly from LLR rather than the network. This is because a network alone is insufficient to handle occlusions effectively, often resulting in significant deviations (Fig. 5). As suggested, we added additional ablation studies on the OpenCAS dataset to investigate the role of activation functions (i.e., ReLU and SIREN) with the LLR for regularization. The results presented in the following table reveal that  the regularization of deformations in occluded areas is primarily attributed to LLR rather than the activation function, as ReLU+CD and ReLU+MCC still generate significant deformation errors, especially for both the EPE and Outlier metrics.
>
> - |Method   |EPE↓|AccS↑| AccR↑|Outlier↓|EPE↓|AccS↑|AccR↑|Outlier↓| EPE↓|AccS↑|AccR↑|Outlier↓|
> |-|--|-|-|-|-|-|-|-|-|-|-|-|
> |ReLU+CD  |47.994|9.106|20.466|17.957|39.171|9.684|23.321|8.533| 27.494|28.766|44.443|11.272|
> |ReLU+MCC|31.582| 28.446| 40.614|   4.378|12.012| 47.258| 69.975| 0.000|17.245| 32.841| 51.744|  1.326|
> |ReLU+MCC+LLR|**14.690**|**34.144**| **57.860**|**0.000**|**6.984**|**66.543**|**89.956**|**0.000**|**14.306**| **36.670**| **52.982**|**0.000**
> - Moreover, to investigate the intrinsic regularization capabilities of activation functions, we have conducted an additional experiment by applying recursive deformations to complete shapes without occlusion. The EPE metric detailed in the subsequent table demonstrates that ReLU and SIREN exhibit comparable performance on complete shapes. This finding further substantiates **our conclusion that while activation functions do possess a regularization effect, they may fall short when addressing the complexities of challenging occlusions (i.e., the motivation and core contributions of this work). In such cases, LLR proves to be particularly beneficial.** We have added these additional ablation studies in Sec. E of the Appendix of the revised manuscript.
>
> - |Method   |Liver 1->Liver 2|Liver 2->Liver 3|Liver 3->Liver 1
> |-|-|-|-|
> |ReLU+CD  |4.685|5.550|6.787|
> |SIREN+CD|3.912| 5.063| 6.183|
>
> - **Differences and Connections with Robust Functions:** Thanks for this insight. We have conducted an in-depth analysis comparing correntropy with robust functions.
>
>   **1) Differences**. Unlike common robust functions, correntropy is initially proposed in information-theoretic learning to handle **nonzero mean and non-Gaussian noise** (e.g., the occlusion part can be seen as a certain type of non-Gaussian noise), which is related to the Renyi’s quadratic entropy. Besides, Maximum correntropy criterion (MCC)  is a **local measure** that provides a **probabilistic meaning** of maximizing the error probability density at the origin according to the information potential.
>
>   **2) Connections**. As suggested, we derive the relationship between MCC and robust functions. By setting $\rho(e)=\left(1-\exp \left(-e^2 / 2 \sigma^2\right)\right) / \sqrt{2 \pi} \sigma$, we can prove that $\rho(e)$ is an influence function satisfying all the criteria of a robust function, i.e.,  1) $\rho(e) \geq 0$; 2) $\rho(0)=0$; 3) $\rho(e)=\rho(-e)$; 4) $\rho\left(e_i\right) \geq \rho\left(e_j\right)$ for $\left|e_i\right|>\left|e_j\right|$. Moreover, $\min_{\theta}\sum_{i=1}^{N}\rho\left(e_{i}\right)\\Leftrightarrow \max_{\theta} \sum_{i=1}^{N} \exp\left(-e_{i}^{2} / 2 \sigma^{2}\right)/ \sqrt{2 \pi} \sigma$ with the corresponding weight function of $\rho(e)$ as $w(e)=\exp \left(-e^2 / 2 \sigma^2\right) / \sqrt{2 \pi} \sigma^3$. For comparison, the Bi-square's weight function is $w_{\mathrm{Bi}}(e)= \left[1-(e / h)^2\right]^2  \quad \text{if} \quad |e| \leq h \\ \quad \text{else}  \quad 0$, where $h$ is the tuning threshold. Therefore, the nonzero part of $w_{\mathrm{Bi}}(e)$ is equivalent to (with a constant) the square of the first-order Taylor expansion of $w(e)$. MCC is analogous to a robust function but with the specific influence function $\rho(e)$. However, unlike the Huber or Bi-square function, correntropy does not need a predefined threshold such as $h$ and the kernel size entirely governs the properties. Moreover, our derivation between correntropy and robust functions offers a practical way to select an appropriate threshold for robust functions or determining the kernel size for correntropy. We added these analyses in Sec. F of the Appendix of the updated manuscript to relate MCC with robust functions.

---

> > ### Comment · Reviewer_Jzhm · 2024-11-27
> > **Importance of MCC still unclear**
> >
> > The authors did a good job of addressing my other concerns around the role of architecture on the regularization, and also the importance of LLR in regularization. In this regard I am satisified with the authors responses.
> >
> > The derivation they show for relating the MCC with the robust functions is useful. However, I really worry about how the paper is presented at the moment in this regard. The role of correntropy (i.e. MCC) is overblown. The title, abstract and introduction of the paper make a significant deal about correntropy. However, by the authors own admission the central utility of the approach is around the LLR regularization.
> >
> > Since MCC is such a significant part of the narrative of the paper it seems inappropriate for this discussion to be hidden in the appendix. My view is that the paper needs a re-write. I suspect a generic robust-error function will work just as well, and there would be little sensitivity to threshold and kernel size selection (removing the central advantages of the MCC measure that the authors have noted). Simply deriving the connection is not enough. The authors are trying to pass MCC off as something novel to the community, which I still remain unconvinced.

---

> > > ### Author Response · Authors · 2024-11-27
> > > **Thank you for your reply**
> > >
> > > We thank the reviewer for the follow-up. We appreciate the discussion, your feedback, and your suggestions.
> > >
> > > **Re: Satisfied with the regularization analysis and robust function derivation**
> > > Thank you very much for your encouraging comment, which is a great appreciation of our efforts.
> > >
> > > **Re: Paper presentation** Thanks. We indeed observe the physically implausible phenomena with many previous  non-rigid deformation approaches. To address this issue, our method features two key contributions:
> > > 1) A correntropy-induced local metric **adaptively** distinguishes between points, effectively preventing collapse, tearing, and other physically implausible outcomes that have been overlooked in previous approaches. While avoiding physically unreasonable phenomena is crucial, it is not sufficient on its own, as the deformation of **occluded parts** remains uncertain. Our method's first step with MCC ensures that shapes remain reasonable.
> > >
> > > 2) Then, we pioneer the use of LLR for non-rigid deformation, ensuring that deformations, even for occluded parts, are physically reasonable. This is a significant advancement over existing methods.
> > >
> > > As presented in the abstract and introduction part, our paper is structured to highlight these two **sequential and holistic** contributions. **The central utility of our method lies in the combination of MCC with LLR**, rather than relying on either technique alone.
> > >
> > > **Re: The MCC importance** Thanks.
> > >
> > > 1) We would like to clarify that we do not claim MCC as a novel metric proposed by us. While MCC is an existing metric, our innovative application of MCC within the framework of implicit neural deformation optimization to ensure physically reasonable deformations is what sets our work apart. Specifically, we utilize MCC to ensure physically plausible deformations, particularly for occluded geometries, which is a novel contribution.
> > >
> > > 2) MCC is initially proposed in information-theoretic learning to handle nonzero mean and non-Gaussian noise. In this light, the occlusion part can be seen as a certain type of non-Gaussian noise. Moreover, MCC is a local measure that provides a **probabilistic meaning** of maximizing the error probability density at the origin based on the information potential. These characteristics indeed differentiate MCC from common robust functions, as has also been demonstrated in our derivation.
> > >
> > > 3) We use MCC as we have conducted a theoretical analysis comparing MCC and CD (a ubiquitous metric in many point cloud analysis tasks), concluding that the standard CD is a special case of MCC. Given the widespread adoption of the CD metric, we also believe that integrating MCC can offer a valuable enhancement across various domains, including point cloud analysis.

---

> ### Author Response · Authors · 2024-11-16
> **Part 2: Detailed Answer and Discussions to Other Concerns**
>
> We give a detailed answer to other comments and sincerely hope that we have addressed your concerns.
>
> # Re: Weaknesses
> ---
> - **Our Motivation and Contributions:** In this work, we develop a **systematic unsupervised and runtime-optimization** non-rigid registration method to address the challenging issue of **occlusions**, a problem that has not received adequate attention in prior approaches. Our method features two key contributions: 1) A **correntropy-induced local metric** that adaptively distinguishes between points, thereby preventing the collapse, tearing, and other physically implausible outcomes encountered in prior methods. Moreover, given the widespread adoption of the Chamfer distance, we believe that the MCC metric can serve as a valuable enhancement across various domains such as for the point cloud anlaysis field. 2) We **pioneer** the use of **local linear reconstruction** for non-rigid registration to ensure **physically reasonable** deformations for occluded parts. Our method demonstrates superior performance over competitors and offers a valuable algorithm for managing occlusions effectively.
>
> - **Discussion of the Suggested Work:** Thanks for the valuable suggestion. A distance transform is utilized to expedite the computation of the Chamfer distance in [1], and in principle, this manner can be extended to the MCC-induced metric as our proposed method also incorporates pairwise computation. However, as the authors noted, we must carefully consider the trade-off between discretization error, grid resolution, memory consumption, and estimation accuracy. We have cited and added these discussions of the recommended work in the revision.
>
>
> # Re: Questions
> ---
> - **Bandwith  in Definition 1:** Yes, Definition 1 is applicable to general Mercer kernels. We use bandwith here as our analysis is centered around the Gaussian kernel. We have updated the manuscript for general Mercer kernels and emphasized the use of Gaussian kernel.
>
> - **Kernel in Lemma 1:** The kernel used in Lemma 1 is also the Gaussian kernel. Moreover, as Gaussian kernel is a translation-invariant kernel, we use $k(x-y)$.
>
> # References
> ---
> [1] Li et al., Fast Neural Scene Flow, ICCV23.

---

> ### Author Response · Authors · 2024-11-24
> **End of Discussion Approaching**
>
> Dear Reviewer Jzhm,
>
> Thank you very much for reviewing our paper. We have carefully and diligently updated the manuscript to reflect the changes mentioned above. We hope our responses have addressed your concerns.
>
> Since it is approaching the end of the discussion period, should you have any further questions, we're happy to discuss and clarify.
>
> We appreciate the discussion, your feedback, and your suggestions. Many thanks for your time and effort.
>
>
> Best and sincere wishes,
>
> The authors

---

### Author Response · Authors · 2024-11-25

Dear reviewers,

Thank you very much for taking the time and efforts to review our paper. We have carefully revised the paper draft to incorporate updates provided in the rebuttal.

If you have any further questions or feedback, please don’t hesitate to let us know!

Best regards,

The Authors

---

### Meta-Review · Area_Chair_jaru · 2024-12-24

**Metareview:**

This paper introduces an approach for non-rigid point cloud registration. The key idea is to utilize implicit deformation specially designed for handling occlusions. The proposed approach works in an unsupervised manner based on an adaptive correntropy function for local similarity measures. The paper extensively validated the non-rigid point registration under severe occlusions.

The summary of the strength is as follows:
- Impressive results
- Good application of local linear reconstruction (LLR) regularization loss
- Use of maximum correntropy criterion (MCC) metric, although it is the existing metric
- Detailed ablation study
- Good presentation and writing

The summary of the weakness of this paper is as follows:
- Limited technical novelty
- Minor mistakes in Lemma and Definitions
- Slow runtime
- Missing comparison with other datasets (demonstrated with synthetic datasets)

In summary, the paper received the scores on the bar. All reviewers do not recommend strong acceptance or rejection of this paper. However, AC notes that the merits introduced in this paper can shed light on non-rigid point cloud registration, with the interesting idea that it enables the training in an unsupervised manner. AC also confirms that the generality of the proposed approach can be widely adopted in various applications. Given these, AC recommends acceptance of this paper.

**Additional Comments On Reviewer Discussion:**

This paper received constructive reviewer feedback, and the authors provided detailed answers.

Specifically, reviewer Jzhm asks about analyzing the regularization term, the difference between correntropy and robust functions, and technical contributions. The authors provided detailed feedback, and the reviewer Jzhm stated that it resolves several concerns. However, the reviewer also noted that the highlight of correntropy in this paper is not appropriate since other robust functions can replace it. AC asked for additional opinions during the Reviewer-AC discussion phase, but the reviewer, Jzhm, did not reply. The score by reviewer Jzhm remains unchanged at 5.

The other reviewers gave a score of 6. The reviewer FdAL requests the ablation study on MCC and Chamfer distance, generalization capability, and additional experiments on mesh hole filling. The authors provided solid feedback with the additional results, and the reviewer states that the questions were adequately addressed and kept a positive rating. The reviewer C5yq requests additional experiments on the SHREC'19 and SHREC'16 datasets, and the reviewer asks about the difference from the prior work. The reviewer stated that the authors' answers are not clear in some aspects (why LLR has such an impact), but the reviewer kept the original score of 6. The reviewer Rxao asks about the incorporation of geometric descriptors, motivation of implicit neural fields, and additional ablation study. The reviewer Rxao mentioned that the authors' feedback resolved the concerns and kept the score of 6.

AC confirms that the discussion phase was constructive, and the many questions raised by reviewers were properly addressed by the authors.

---

### Decision · Program_Chairs · 2025-01-22

Accept (Poster)